# Avian-inspired embodied perception in biohybrid flapping-wing robotics

Qian Li [1], Ting Tan [2] ✉, Benlong Wang [1] & Zhimiao Yan [1] ✉

Avian feather intricate adaptable architecture to wing deformations has catalyzed interest in feathered flapping-wing aircraft with high maneuverability, agility, and stealth. Yet, to mimic avian integrated somatic sensation within stringent weight constraints, remains challenging. Here, we propose an avian-inspired embodied perception approach for biohybrid flapping-wing robots. Our feather-piezoelectric mechanoreceptor leverages feather-based vibration structures and flexible piezoelectric materials to refine and augment mechanoreception via coupled oscillator interactions and robust microstructure adhesion. Utilizing convolutional neural networks with the grey wolf optimizer, we develop tactile perception of airflow velocity and wing flapping frequency proprioception. This method also senses pitch angle via airflow direction and detects wing morphology through feather collisions. Our low-weight, accurate perception of flapping-wing robot flight states is validated by motion capture. This investigation constructs a biomechanically integrated embodied perception system in flapping-wing robots, which holds significant promise in reflex-based control of complex flight maneuvers and natural bird flight surveillance.

The exceptional agility of avian species owes much to their multi-degree-of-freedom wing structure. By adjusting wing movements, birds achieve complex aerial behaviors[1], like efficient cruising, nimble maneuvering, and precise descent. Besides the flexible wing skeleton, the unique arrangement of feathers forming a distinct wing surface structure plays a pivotal role in avian flight, providing significant aerodynamic advantages[2–4]. Feathers, as natural materials, offer both the necessary strength to uphold lift and the flexibility for seamless overlapping with neighboring feathers. In addition, minuscule barbules on feather surfaces enhance friction, ensuring secure interlocking during wing extension[5]. Utilizing feathers as discrete units enables seamless coordination with muscles, facilitating wing deformation for posture adjustment. For example, hummingbirds proficiently execute various flight maneuvers, including rotation, hovering, and turning[6–8]. Swallows exhibit precise control over descent and turning rates by adjusting the orientation of their posterior wings[9].

The distinct wing structure of avian feathers has sparked interest in designing biomimetic aircraft. Recent concepts propose variable-wing aircraft inspired by birds, using discrete feathers in their construction[10–14]. Compared to fixed-wing designs, wing deformation can extend the gliding distance by 60% and increase gliding time by 100%. Using discrete feathers addresses issues like wrinkling in membrane wings during contraction. In addition, these adaptable feathers significantly enhance aircraft maneuverability and agility, allowing faster and tighter turns[9] and enhancing flight concealment. These studies not only provide fresh insights for the advancement of future unmanned aerial vehicles but also deepen our understanding of the fundamental principles guiding avian flight[15]. The concept of discrete deformable wings holds the potential for achieving more intricate flight behaviors, highlighting the need for a correspondingly precise and responsive sensory system[16,17].

To establish precise somatosensory perception for a feathered flapping-wing vehicle, we examine avian perceptual systems. Avian

[1]State Key Laboratory of Ocean Engineering, Department of Engineering Mechanics, School of Ocean and Civil Engineering, Shanghai Jiao Tong University, Shanghai, China. [2]State Key Laboratory of Mechanical System and Vibration, School of Mechanical Engineering, Shanghai Jiao Tong University, Shanghai, China. ✉e-mail: tingtan@sjtu.edu.cn; zhimiaoy@sjtu.edu.cn

movements and stable flight rely on tactile and kinesthetic sensory systems linked to their wings. Birds detect momentary pressure and vibrations through wing tactile sensors. For instance, Herbst corpuscles enable sensing vibration frequency and displacement amplitude via feathers, aiding in flow velocity perception and assessments of stall and turbulence[18]. Kinesthetic perception, relying on muscle spindles and joint receptors, detects changes in muscle tension, pressure, and joint extension[19,20]. These receptors transform sensory changes into neural impulses, regulating skeletal muscle movement via the somatosensory motor cortex, vital for perceiving body position, posture, and spatial movement adaptations. Despite distinct roles, research indicates both tactile and proprioceptive senses perceive cellular tension through the Piezo2 mechanosensitive protein[21,22], revealing a shared basis for touch and proprioception. Lightweight, sensitive, and adherent flexible Polyvinylidene Fluoride (PVDF) piezoelectric films show promise in effective reflection of pressure and vibration behaviors[23–26]. Integrating multiple sensory perceptions using functional materials and expanding multifunctional structure development is crucial for flapping-wing robots (FWRs)[27,28], especially considering stringent payload weight limitations. Presently, flexible sensing in unmanned aerial vehicles mainly focuses on fixed-wing configurations[29–33]. For flapping-wing aircraft, which have more complex power principles and stricter weight limitations, flexible embodied perception in actual flight remains challenging[34–36].

Organisms in nature serve as a rich source of inspiration for developing functional structure and material[37–39]. Feathers possess a distinctive structure and composition that grants avian species inherent advantages in self-perception[40,41]. Research indicates that most feathers exhibit distinct vibration modes and frequencies under varying flow velocities and directions. The adjacent vibratory feathers might operate as coupled oscillators, culminating in collisions that engender nonlinear interactions[42]. Feathers exhibit remarkable responsiveness to external airflow while boasting lightweight, tear-resistant characteristics[43,44]. Moreover, they possess a

restitution property, enabling their surface morphology to be restored following disorder, merely through the application of gentle stroking or tidying.

In this work, drawing upon natural feather biomaterials and structures to enhance the differentiation of mechanical perception, integrating flexible lightweight self-powered PVDF to emulate the functions of avian wing mechanoreceptors, we propose a biohybrid embodied perception approach with deep learning (Fig. 1a and Supplementary Movie 1). The investigation focuses on the heterogeneous interface connection properties of the feather-PVDF biohybrid mechanoreceptor, encompassing characteristics like peeling strength, fatigue durability, and electromechanical conversion efficiency. Based on avian-inspired flapping-wing robots (FWR) with feathered wings, we achieve a biohybrid tactile and kinesthetic system with real-time classification and visualization of flapping frequency, wind speed, pitch angle, and wing shape (Fig. 1b). In addition, we complete accurate perception of flapping frequency and relative flow velocity during untethered flight. This biohybrid embodied perception design holds the potential for advancing the development of more lightweight, integrated, stealthy, and dexterous biomimetic flapping-wing air vehicles.

## Results
### Biohybrid mechanoreceptor
Analogous to mechanoreceptors in the skin of vertebrates that perceive surface pressure and vibrations, the PVDF piezoelectric material also exhibits rapid and robust responses to pressure and vibrations (Supplementary Fig. 1). The feather-PVDF biohybrid mechanoreceptor is designed to perceive subtle vibrations and surface loads through feathers. Simultaneously, it achieves force-to-electric conversion through the piezoelectric material that adheres conformationally to the feathers. This process resembles the generation of electrical signals by sensory neurons in response to stimulation. Consequently, the conformational co-adhesion of the piezoelectric material with the

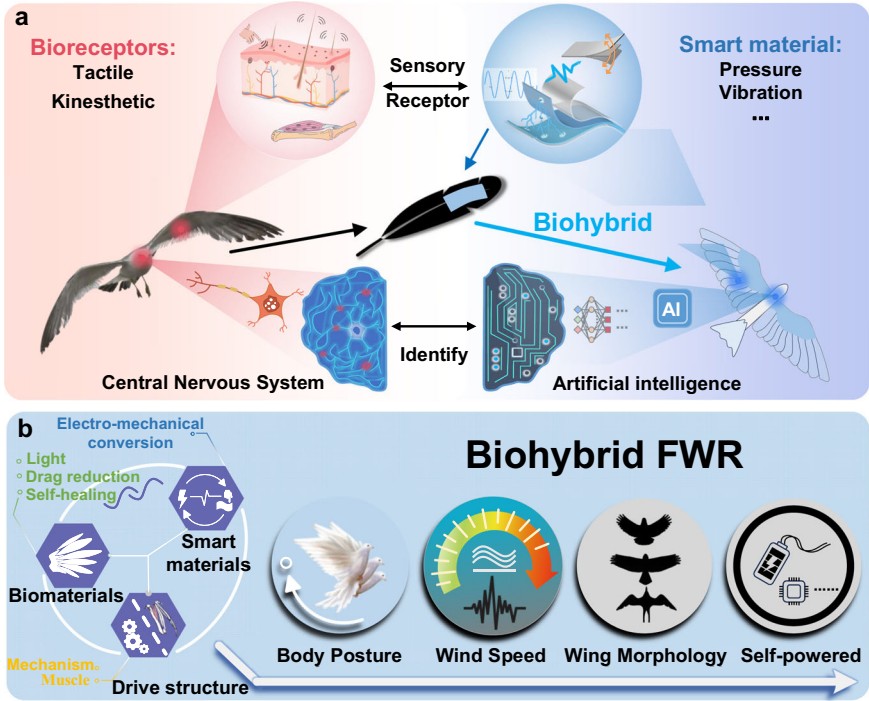

**Fig. 1 | Design schematic diagram of biohybrid flapping-wing robot system.**
**a** Analogous to avian sensors, flexible smart materials can respond to vibrations, pressure, and other stimuli. Inspired by avian tactile and kinesthetic perception systems, we propose a feather-smart material biohybrid mechanoreceptor, achieving recognition of the environment and motion through artificial intelligence algorithms. **b** Biohybrid flapping-wing robots (FWR) composed of biomaterials, smart materials and structures with embodied perception of body posture, wind speed, wing morphology, and embodied energy design.

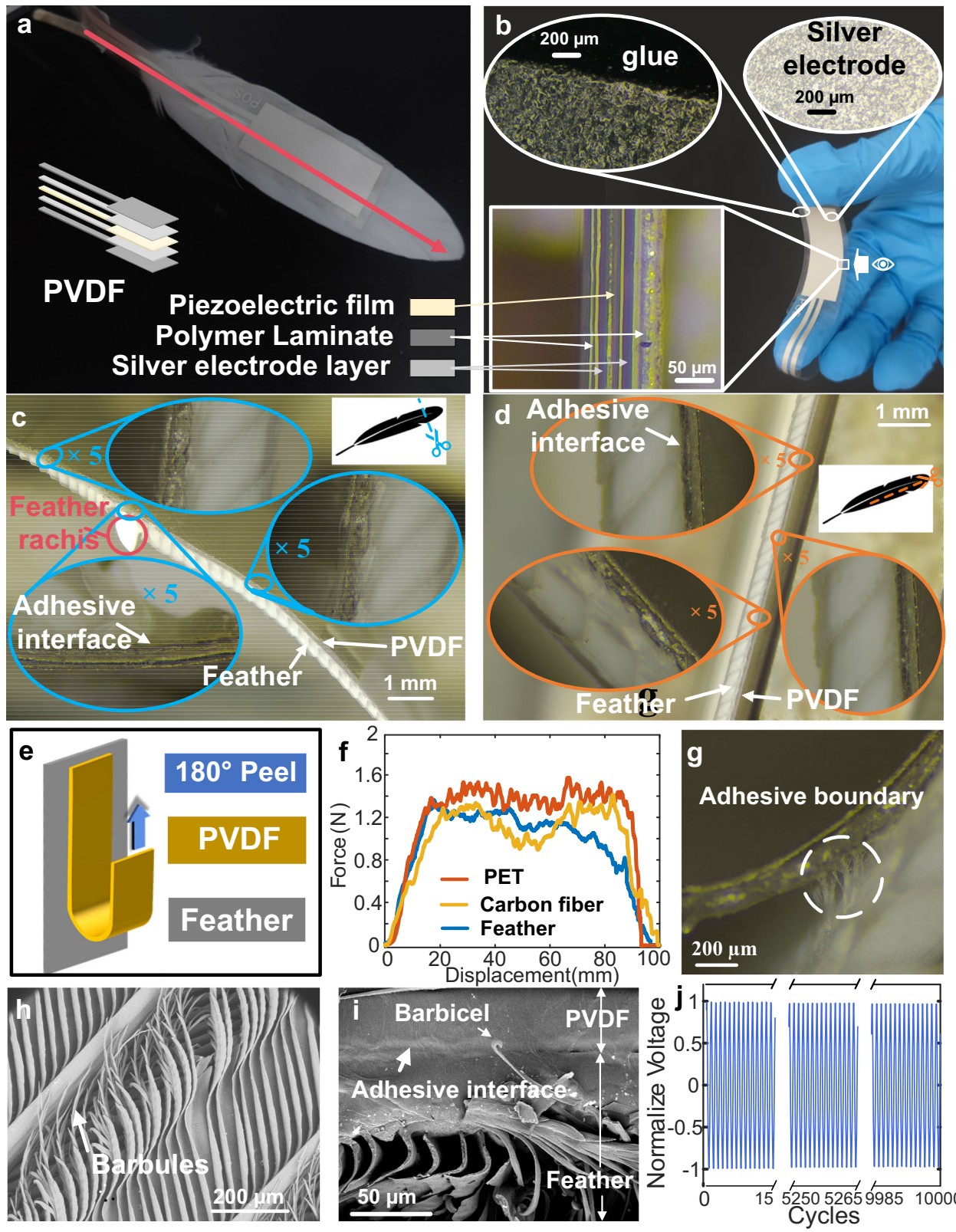

feathers is critical for the functionality of the feather-PVDF biohybrid mechanoreceptor.

Avian feathers typically comprise four components: the rachis, barbs, barbules, and barbicels. The rachis bears numerous parallel barbs on both sides, each barb densely adorned with barbules equipped with tiny hooks. These adjacent barbules interlock, uniting the barbs into a smooth vane (Supplementary Fig. 2). We collected blade-

shaped remiges from geese. In the midsection of the vane, PVDF piezoelectric films are affixed along the rachis direction (Fig. 2a, material characterization in Supplementary Fig. 3). The total thickness of the PVDF film is 125 μm, featuring a multilayered structure consisting of a polymer layer, piezoelectric layer, and electrode layer (Fig. 2b). The uniform distribution of tiny silver particles on the electrode surface ensures excellent conductivity of the PVDF piezoelectric film. A

**Fig. 2 | Characterization and performance testing of feather-PVDF biohybrid mechanoreceptor. a** Feather-PVDF biohybrid mechanoreceptor, including a schematic diagram of PVDF's multilayer structure, the red arrow represents the direction of the rachis. **b** Microscopic characterization of PVDF multilayer structure, electrode layer, and adhesive surface under an optical microscope. **c** Microscopic characterization for the adhesive cross-section of feather-PVDF along the blue dashed line perpendicular to the feather rachis in the upper right corner of the figure. **d** Microscopic characterization for the adhesive cross-section of feather-PVDF along the orange dashed line parallel to the feather rachis in the upper right corner of the figure. **e** Schematic of the peeling test for feather-PVDF. **f** Peeling force curves of feather-PVDF (blue), PET-PVDF (red), and carbon fiber-PVDF (yellow) structures as a function of peeling length. Source data are provided as a Source Data file. **g** Microscopic structure of the bonding interface during the peeling process of feather-PVDF. **h** Microstructure of the feather surface. **i** Microstructure of the heterogeneous attachment cross-section between the feather and PVDF. **j** Normalized voltage curves of feather-PVDF biohybrid mechanoreceptor with the number of vibrations. Source data are provided as a Source Data file.

homogeneous adhesive microstructure is observed on the PVDF adhesive surface (Fig. 2b), promoting effective heterogeneous interface bonding. Feather-PVDF composite materials are sectioned both transversely and longitudinally along the feather, and cross-sections are examined at three random locations in each direction (Fig. 2c, d). Close adhesion between PVDF and feathers is consistently observed in both directions without obvious gaps at the heterogeneous interface.

Peeling tests on feather-PVDF composite material were performed to assess the adhesive mechanical properties of the biohybrid mechanoreceptor (Fig. 2e and Supplementary Movie 2), with specific procedures outlined in the methods. As Polyethylene terephthalate (PET) and carbon fiber are commonly used in aircraft wing structures, they were chosen as control groups for the feather in peel tests. The comparison of peeling force and peeling length curves (Fig. 2f) reveals distinct serrated patterns in the peeling curves for PET and carbon fiber, indicative of adhesive slip oscillations during peeling. In contrast, the PVDF peeling force for feathers exhibits less fluctuation, suggesting a relatively stable peeling process. The average peeling forces are 1.382 N for PVDF-PET, 1.162 N for PVDF-carbon fiber, and 1.122 N for PVDF-feathers, with the latter comparable to the former two. Extensive entwining of adhesive fibers with feather barbs is observed on the peeling surface of PVDF-feather (Fig. 2g). PET and carbon fiber exhibit smoother continuous surfaces, while feathers are covered in fine discrete barbs (Fig. 2h). Similar fibrous features are commonly found in natural biological structures, such as the foot pads of geckos[45]. The fibrous features contribute to increased adhesion in contact with the film. At the micron scale, we observed the cross-section of the heterogeneous interface between the feather and PVDF (Fig. 2i and Supplementary Fig. 4), which exhibited tight adhesion between the interface and interlocking by barbicels of barbules.

To examine the long-time performance of the feather-PVDF biohybrid mechanoreceptor, dynamic fatigue testing was conducted. A mass block was added to the leading edge of a single-wing structure to increase feather deformation amplitude (Methods and Supplementary Movie 3). Normalized voltage curves (Fig. 2j) show that during 10,000 vibration tests, the structural output performance exhibits no significant change. Analysis of local signals and spectra at different stages (Supplementary Fig. 5) display evident single-period characteristics, demonstrating the durable electromechanical conversion performance of the feather-PVDF biohybrid mechanoreceptor even after prolonged fatigue testing.

Natural birds retract their wings during the upstroke to reduce the aerodynamic surface area and minimize drag, and extend their wings during the downstroke to increase the surface area for generating greater lift[46]. This necessitates birds to possess stronger aerodynamic load-bearing capacity during the downstroke. To ensure a similar load-bearing characteristic, we conducted quasi-static bending experiments on a feather-PVDF biohybrid mechanoreceptor (Methods and Supplementary Movie 4). With the feather tip fixed and maintaining equal force arms, single-feather and multi-feather wing structures with PVDF films attached longitudinally on the top surface were individually bent in both clockwise and counterclockwise directions (Fig. 3a). The torque-bending angle curve illustrates that the presence of the PVDF film further enhances the bending load-bearing capacity of the feather structure (Fig. 3b). At the same bending angle, a single feather with

PVDF exhibits larger positive (bending angle > 0°, shown in pink blocks in Fig. 3b) and negative (bending angle < 0°, shown in purple blocks in Fig. 3b) maximum torques compared to a single feather without PVDF. The critical bending angle for the maximum torque in the negative direction increases. The enhancement effect on the negative maximum torque during the downstroke motion of a single feather (2.5 times) is more pronounced than the enhancement effect on the positive maximum torque during the upstroke motion (1.3 times). For multi-feather wing structures, the attachment of piezoelectric films similarly strengthens the bending torque. Although this enhancement is relatively diminished compared to a single feather, it remains advantageous for the load-bearing capacity of the wings. Consequently, the top-surface longitudinal arrangement of the piezoelectric material in this study further improves the load-bearing capacity of the feathered wings against aerodynamic loads during the downstroke motion.

We designed a drive mechanism composed of a gear reduction unit and a DC motor (Supplementary Figs. 6, 7, and Supplementary Movie 1) to achieve flapping motion in single-feather and multi-feather wing structures with attached PVDF films. Utilizing a wind tunnel to simulate flight airflow conditions (Fig. 3c and Supplementary Figs. 8 and 9), we measured the root mean square (RMS) voltage of PVDF at various wind speeds (Fig. 3d) and flapping frequencies (Fig. 3e). With increasing wind speed (0.5-3 m s⁻¹), the RMS voltage generated by both structures exhibited distinct patterns. The single feather structure displayed a trend of initially increasing and then decreasing RMS voltage with rising wind speed, reaching its maximum at a wind speed of 2 m s⁻¹. In contrast, the multi-feather wing structure showed an overall increasing trend in RMS voltage, with significant differences at various wind speeds, making it more conducive to wind speed recognition. As the flapping frequency increased (1−6 Hz), both the single and multi-feather wing structures demonstrated an increase in RMS voltage, following a similar pattern. Although the voltage generated by the multi-feather wing at low frequencies (1−3 Hz) was comparable to that of the single feather wing, at higher frequencies (4−6 Hz), the multi-feather wing produced a more significant voltage enhancement. At a flapping frequency of 6 Hz, the voltage of the multi-feather wing approached 1.5 times that of the single-feather wing. Therefore, the voltage signals of the multi-feather wing structure under different flight conditions of wind speed and flapping frequency exhibit good distinctiveness and regularity, laying a material and structural foundation for the biohybrid perception of touch and motion in avian flapping flight.

## Somatic sensation

Somatic sensation plays a crucial role in the control and regulation of an organism's trunk and limbs. As integral components of somatic sensation[47], both tactile and proprioceptive feedback are essential. Tactile perception endows organisms with the ability to sense external pressures, vibrations, and temperature-humidity variations, while proprioception provides information about body movement and position. These two sensory modalities, distinct yet intricately interconnected, jointly regulate the organism's perceptual system[48,49]. This also provides inspiration for robot perception[50]. The feather-PVDF biohybrid mechanoreceptor serves as a highly adaptable interface with

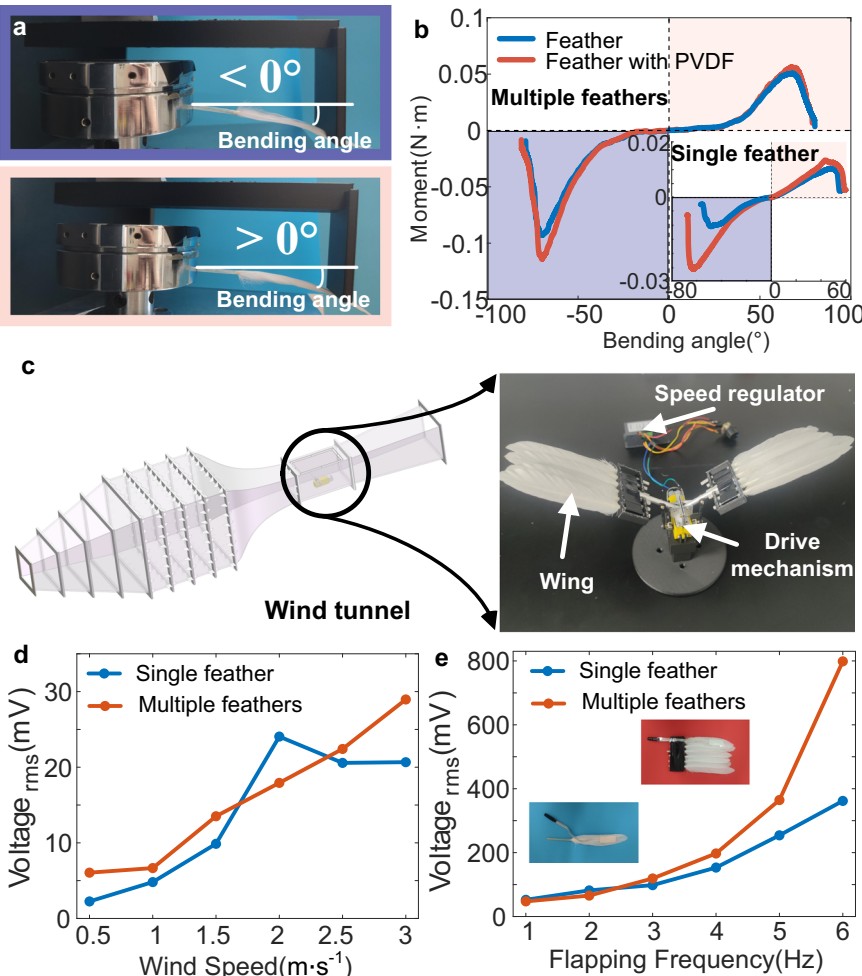

**Fig. 3 | Mechanical and electrical performance testing of feather-PVDF biohybrid mechanoreceptor. a** Schematic diagram of the feather bending experiment apparatus (bending angle < 0°, shown in purple block; bending angle > 0°, shown in pink block). **b** Torque-deflection curves for single feather and multi-feather wing structures with and without attached piezoelectric patches. Source data are provided as a Source Data file. **c** Schematic diagram of the wind tunnel setup and flapping mechanism. **d** Root mean square of voltage signals generated by single feather and multi-feather wing structures with attached piezoelectric patches at wind speeds ranging from 0.5 m s⁻¹ to 3 m s⁻¹. Source data are provided as a Source Data file. **e** Root mean square of voltage signals generated by single feather and multi-feather wing structures with attached piezoelectric patches at flapping frequencies ranging from 1 Hz to 6 Hz. Source data are provided as a Source Data file.

flying objects. It functions as a sensory nerve, providing feedback on touch and movement to facilitate environmental monitoring and recognition of the aircraft's motion state through flight energy recovery (Fig. 4a). The lightweight nature of PVDF effectively reduces the energy burden and payload of onboard sensors. The electrical energy generated during the flapping process can power environmental sensors, such as temperature and humidity sensors (Fig. 4b). Simultaneously, this energy can drive an electronic clock to generate clock signals. Harvesting energy accumulated from flapping motion can also provide power for portable airborne environmental monitoring modules such as light intensity and ultraviolet radiation sensors[51], enhancing the diversity of environmental parameter monitoring and data collection. Although our current energy harvesting efficiency is not remarkable (Supplementary Fig. 10), future improvements can be achieved by optimizing the flapping wing structure and increasing the flapping frequency, providing valuable insights into the energy supply of micro-sized FWRs.

We systematically examined the somatic sensation under different touch environments and body movements, considering flapping frequency, wind speed, and pitch angle (Fig. 4c and Supplementary Movie 5). The feather-PVDF biohybrid mechanoreceptor, acting as the mediator for piezoelectric conversion, demonstrated varied output in electrical voltage signals due to distinct force scenarios in each condition. We employed a quasi-steady flapping wing model[52] that accounts for translational forces induced by leading-edge vortices and rotational forces induced by wing rotation. The nonlinearity introduced by the flow-wing interaction generates additional forces during combined translational and rotational wing movements, explained through a coupling effect. Moreover, the reciprocating wing motion induces airflow acceleration and deceleration, and we computed this force by considering added mass effects. Finally, by superimposing these four force components, we calculated the resultant force on the vertical wing surface and the lift force during flapping motion, offering a qualitative and interpretable explanation for the different outputs in voltage signals.

The perception of wing flapping in birds and aircraft falls under proprioceptive sensation. The periodicity of flapping can be transmitted to mechanical receptors through the wing structure. Flapping motion, as the primary excitation source for the mechanical sensor, results in voltage waveforms with evident and consistent periodicity at the same flapping frequency. However, at different flapping frequencies, the biohybrid mechanoreceptor produces signals with varying amplitudes, aligning with the force distribution on the wing surface (Supplementary Figs. 17, 18a). As the flapping frequency

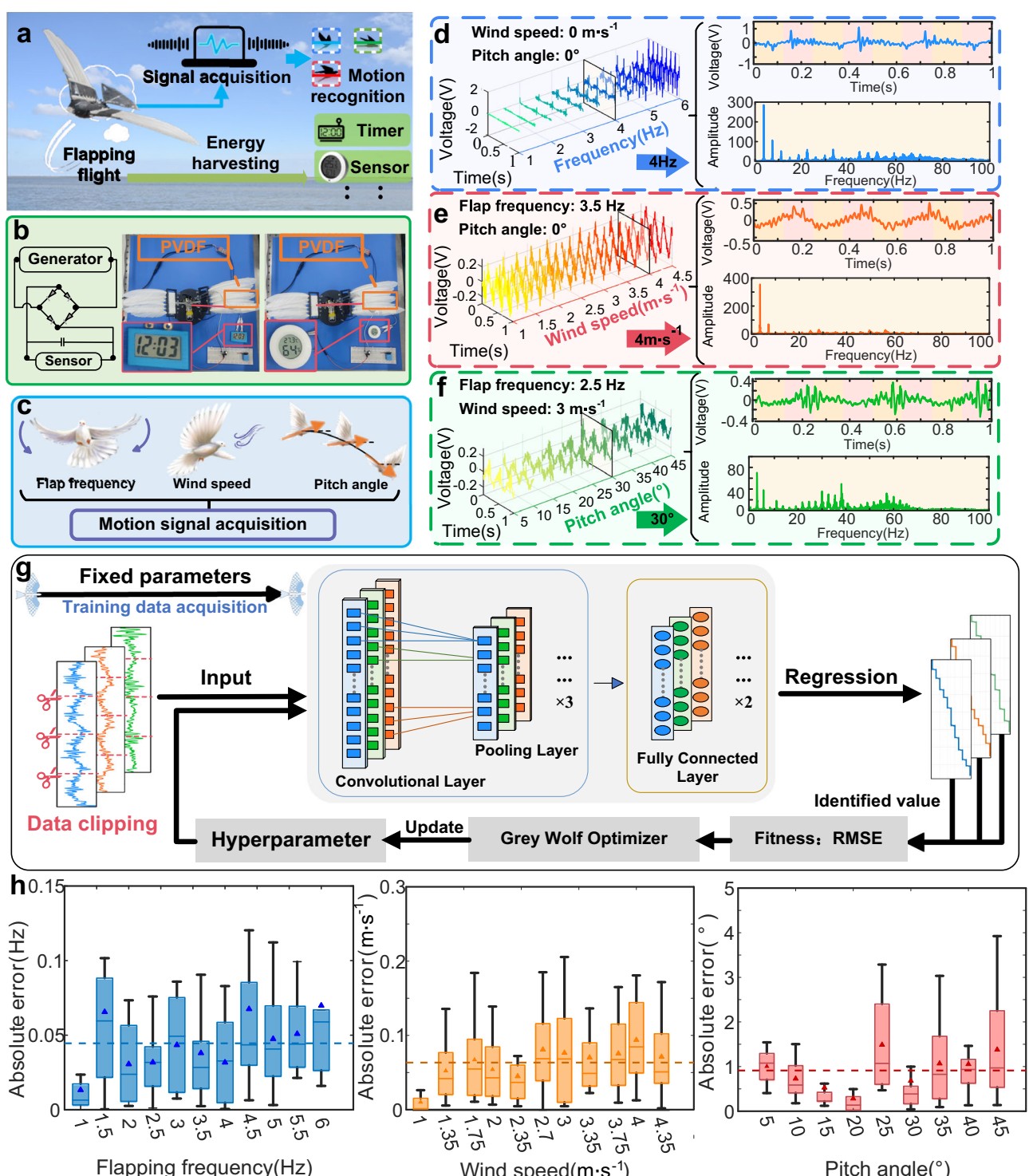

increases, the total force and lift amplitude on the wing surface gradually increase. In the voltage spectrum of a single feather flapping (Supplementary Fig. 11), fundamental frequency and its harmonics indicate the distinctive non-linear characteristics of the feather-flapping system, providing signal features across multiple frequency bands. In addition, due to the presence of numerous thin and flexible barbs forming the feather, irregular vibrations of the barbs during flapping and airflow result in widely distributed noise frequencies (Supplementary Figs. 11–14). Analyzing the spectrum during flapping of a multi-feather wing structure (Fig. 4d and Supplementary Fig. 14), at lower flapping frequencies, fundamental frequency and initial harmonics are noticeable, and chaotic and

prominent frequency components exist around 20–80 Hz. The collisions between feathers in the multi-feather wing structure, especially at lower excitation frequencies, significantly influence the generation of high-frequency signals. With increasing flapping frequencies, the fundamental frequency and its harmonics become more prominent, and the amplitude of the frequency increases. During this phase, the voltage signal is primarily influenced by the flapping motion, while the impact of collisions and vibrations among feathers is relatively diminished.

Perception of wind speed around birds and aircraft involves tactile sensing of oncoming airflow. The airflow directly contacts the wing mechanical receptor, transmitting pressure and vibration to

**Fig. 4 | Feather-PVDF biohybrid mechanoreceptor for embodied energy and voltage signal acquisition, and convolutional neural network-gray wolf optimizer algorithm for embodied perception recognition. a** Voltage generated by smart materials during flight in different environments (flapping frequency, wind speed, pitch angle) for environmental and motion recognition, as well as energy harvesting. **b** Demonstration of embodied energy harvesting, including the charge-discharge circuit diagram driving clock and temperature-humidity sensor. **c** Three control variables for motion signal acquisition: flapping frequency, wind speed, and pitch angle. **d** Voltage time-history curves at 1–6 Hz flapping frequency with 0 m s$^{-1}$ wind speed and 0° pitch angle, showcasing the example at 4 Hz flapping frequency and its corresponding voltage time-history curve and spectrum. Source data are provided as a Source Data file. **e** Voltage time-history curves at 3.5 Hz flapping frequency with 0° pitch angle and wind speeds ranging from 1 to 4.35 m s$^{-1}$,

showcasing the example at 4 m s$^{-1}$ wind speed and its corresponding voltage time-history curve and spectrum. Source data are provided as a Source Data file. **f** Voltage time-history curves at 2.5 Hz flapping frequency with 3 m s$^{-1}$ wind speed and pitch angle ranging from 5° to 45°, showcasing the example at 30° pitch angle and its corresponding voltage time-history curve and spectrum. Source data are provided as a Source Data file. **g** The workflow of convolutional neural network (CNN) and gray wolf optimization (GWO) algorithm. **h** Box plots of the absolute errors in the identification of flapping frequency (blue, number of samples $n = 132$), wind speed (yellow, number of samples $n = 132$), and pitch angle (red, number of samples $n = 108$). The box plots show the maximum, minimum, 25th percentile, 75th percentile, and median error values for each category. Triangular points represent the mean error for each category, while the dashed line indicates the mean error across all test samples. Source data are provided as a Source Data file.

structural units. While the airflow-induced voltage on feathers, distinct from flapping, exhibits a minor overall amplitude (Supplementary Figs. 12, 13), due to the nonlinear effects (Supplementary Figs. 19–23) of the hybrid mechanoreceptor, different wind speeds produce complex vibration signals of varying frequency bands and amplitudes, resulting in recognizable feature differences. This can be utilized to achieve tactile perception by the wings of the oncoming airflow. With a fixed flapping frequency, as the wind speed increases, the signal maintains the same periodicity, with minimal changes in the fundamental frequency amplitude, gradual reduction of the harmonic components, and diminishing noise at high frequencies (Fig. 4e and Supplementary Fig. 15). This suggests that at low wind speeds, the multi-feather wing structure undergoes more complex vibrations. Perception of the pitch angle in birds and aircraft pertains to tactile sensing of the oncoming flow direction. At different wind pitch angles, the airflow exerts different aerodynamic forces on the wings. With an increasing pitch angle, the overall force-time curve of the wing surface gradually decreases, while the peak-to-peak value remains essentially constant, and lift exhibits the same pattern as the total force curve (Supplementary Figs. 17, 18b). Observing the voltage signals at different pitch angles with fixed frequency and wind speed (Fig. 4f and Supplementary Fig. 16), changes in the pitch angle result in varying amplitudes of harmonic components and noise distribution.

Differing from the conventional approach, where various types of sensors are responsible for distinct perceptual functions, the biohybrid perceptual method integrates tactile and proprioceptive perception, achieving proprioception through the tactile modality[48]. This enhances the integrative nature of multisensory perception, further reducing the payload of winged robots. Upon receiving tactile and proprioceptive information, the avian mechanoreceptor generates electrical signals through neurons, transmitting them to the brain for cognitive processing. Analogous to the working mechanism of the central nervous system, convolutional neural networks exhibit exceptional feature extraction capabilities[53]. Compared to fully connected feedforward networks[30,33], one-dimensional convolutional neural networks (1D CNNs) utilize local connections and parameter sharing, resulting in lower computational complexity and reduced risk of overfitting, demonstrated in applications such as electrocardiogram monitoring[54] and structural damage detection[55,56]. The Gray Wolf Optimizer (GWO), inspired by the natural behaviors of gray wolf packs[57], can reliably, quickly, and efficiently explore the search space, making it widely used in optimizing neural network hyperparameters. Three separate 1D CNNs were trained for the extraction of corresponding piezoelectric signal data features and regression identification of the three flight parameters: flapping frequency, wind speed, and pitch angle. We employed GWO to optimize critical parameters for each CNN network (Supplementary Tables 1–3 and Supplementary Movie 6). To balance recognition accuracy with the complexity of network training, we choose 1 second as the optimal size (Supplementary Fig. 24) of the input

sample. The schematic diagram of the data extraction and identification process is depicted in Fig. 4g. We calculated the absolute errors and data distributions in the test set (Fig. 4h), yielding an average absolute error of 0.043 Hz for flapping frequency perception, 0.064 m s$^{-1}$ for wind speed perception, and 0.910° for pitch angle perception. Despite occasional outliers in absolute errors for specific categories, the overall average error remains at a low level. The relative error distributions and iteration process for the identification of the three parameters are shown in Supplementary Figs. 25 and 26, indicating excellent regression fitting accuracy of the trained networks. We contrast the perception errors of our flapping-wing flight against fixed-wing flight[33,58–61] (Supplementary Table 4), evidencing either equivalent or elevated recognition fidelity, especially in the context of velocity identification. It is noteworthy to point out that our approach utilizes a single bio-hybrid sensor, effectuating a notable diminution in both complexity and weight for the sensor assembly.

Given the highly maneuverable nature of avian flight, prolonged flight often necessitates modulation of the flapping frequency to accommodate environmental conditions and operational demands (Fig. 5a). To assess the perceptual recognition performance of the trained neural network for variable-frequency flapping motion over an extended period, we employed Arduino and L298N motion module to control the wing flapping frequency for collecting voltage signals (Supplementary Fig. 27) during variable-frequency flapping motion. The experimental setup is depicted in Fig. 5b, and the circuit connection is illustrated in Fig. 5c. The specific experimental procedure is detailed in the Methods. To ensure sufficient temporal resolution during the recognition process, we employed a sliding window approach to segment variable-frequency time-series signals (Fig. 5a). Subsequently, neural networks in Fig. 4g were utilized to identify changes in flapping frequency and generate fitted curves of flapping frequency over time. Results of testing and recognition under a sampling sliding step of 0.01 s are shown in Fig. 5d, e (Supplementary Movie 6 for data processing details). Comparing results under different sliding step sampling rates (Supplementary Fig. 28) reveals that smaller steps lead to denser recognition points and higher temporal resolution during the process of flapping frequency variation over time. This effect is particularly pronounced in intervals with high rates of flapping frequency change, highlighting the enhanced temporal resolution achieved by reducing the sliding step. In addition, this approach mitigates output latency introduced by sampling, ensuring accurate and real-time recognition. Based on this method, we provided a comparison between test and recognition values for two frequency variation patterns (Fig. 5f). The data points are concentrated around the black dashed line with a slope of 1, indicating a high level of agreement between the recognition and test results. The average absolute errors are 0.0722 Hz (sin mode) and 0.0866 Hz (step mode), respectively. This indicates that the trained CNN exhibits a significant predictive effect on time-varying frequency signals with excellent recognition accuracy.

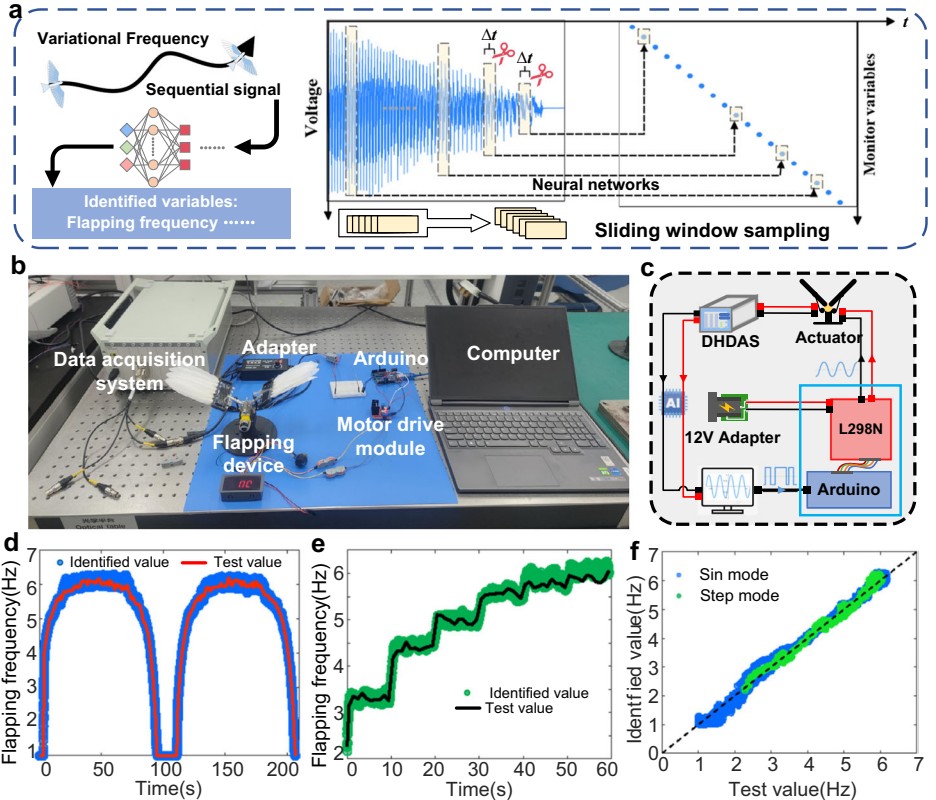

**Fig. 5 | Experiment of frequency-varying motion signal recognition in continuous time. a** Schematic diagram of the recognition of continuous-time frequency-varying motion signals. A sequential voltage signal is segmented with a time interval of Δt = 1, input into the network for prediction, and finally, the corresponding predicted variable curve for this time series is obtained. **b** Physical connection diagram of the experimental setup, including the data acquisition system (DHDAS), computer, power supply, flapping motor, Arduino microcontroller, and L298N drive module. **c** Schematic diagram of the experimental setup connection circuit. **d** Time-history curves of the flapping frequency with a sinusoidal variation in continuous time, showing the test values and predicted values (sliding step = 0.01 s, number of samples n = 20700). Source data are provided as a Source Data file. **e** Time-history curves of flapping frequency with a stepwise variation in continuous time, showing the test values and predicted values (sliding step = 0.01 s, number of samples n = 5941). Source data are provided as a Source Data file. **f** Comparison of test values and identified values of flapping frequency under the two variation modes in (**d** and **e**). Source data are provided as a Source Data file.

## Real-time embodied perception

Building upon the recognition of biological hybrid perception through experimental data from flapping mechanisms, we applied deep transfer learning to motion recognition on a bird-like feathered flapping robot (Supplementary Fig. 29) to achieve embodied perception. The feathered flapping robot adapted from the Meta-Bird model (Bionic Bird) features a comprehensive driving and control system powered by a 1.6-watt micro-coreless motor and LI-PO battery, with a flapping frequency ranging from 0 to 9 Hz. High-speed cameras captured the flapping motion of the bird-like feathered flapping robot during one complete cycle, as illustrated in Fig. 6a. By controlling the throttle of the robot, allowing it to approximately linearly increase from 0% to 100% over a specific time, covering the range of the robot's minimum and maximum flapping frequencies (Supplementary Movie 5), the collected voltage time series is depicted in Fig. 6b. This data was segmented into samples and used as input for deep transfer learning. We retained certain structural parameters trained from the source data in the original CNN, including convolutional layers, pooling layers, and activation functions, updating only the parameters of the fully connected layers to obtain a new neural network (Fig. 6d). It is noteworthy that this process did not involve extensive signal collection under different conditions as shown in Fig. 4b; instead, it relied solely on the time series signal shown in Fig. 6b for transfer training. The new network was then employed to recognize a segment of new approximately linearly varying frequency signals (not involved in the transfer learning training) and a segment of irregularly varying

frequency signals (Fig. 6c and Supplementary Movie 7). The recognition results (sliding step = 0.01 s) are presented in Fig. 6e. The average errors between predicted and tested values (Supplementary Fig. 30) were 0.21 and 0.19, respectively. In addition, we compared the recognition results obtained by the original network without transfer training (Supplementary Fig. 31), with average errors of 0.31 and 0.46, respectively. This indicates that through deep transfer learning, we can quickly fine-tune the original neural network with less data and faster training speed, enabling rapid adaptation to different motion characteristics of the robot achieving motion perception functionality without the need for extensive re-collection of data.

Besides flapping frequency, birds alter the morphology of their wings to enhance maneuverability and agility, adapting to the flight environment[9,62]. Multi-feathered wings consist of discrete feathers, allowing for convenient modification of wing morphology. The perception of wing morphology falls under proprioception, requiring the avian robot to be aware of the overall arrangement of feathers for feedback adjustments. Emulating the distribution shape of feathers at the leading edge of bird wings and varying the relative positions (angle, spacing) between feathers, we fabricated three wing morphologies (elliptical wing, high-lift wing, and soaring wing) and mounted them on the bird-like flapping robot (Fig. 6f). Different wing morphologies exhibit distinct feather arrangements, resulting in varied forms of feather collisions during flapping. Therefore, utilizing collision tactile perception between feather units to achieve proprioceptive perception of wing morphology demonstrates the homologous fusion of

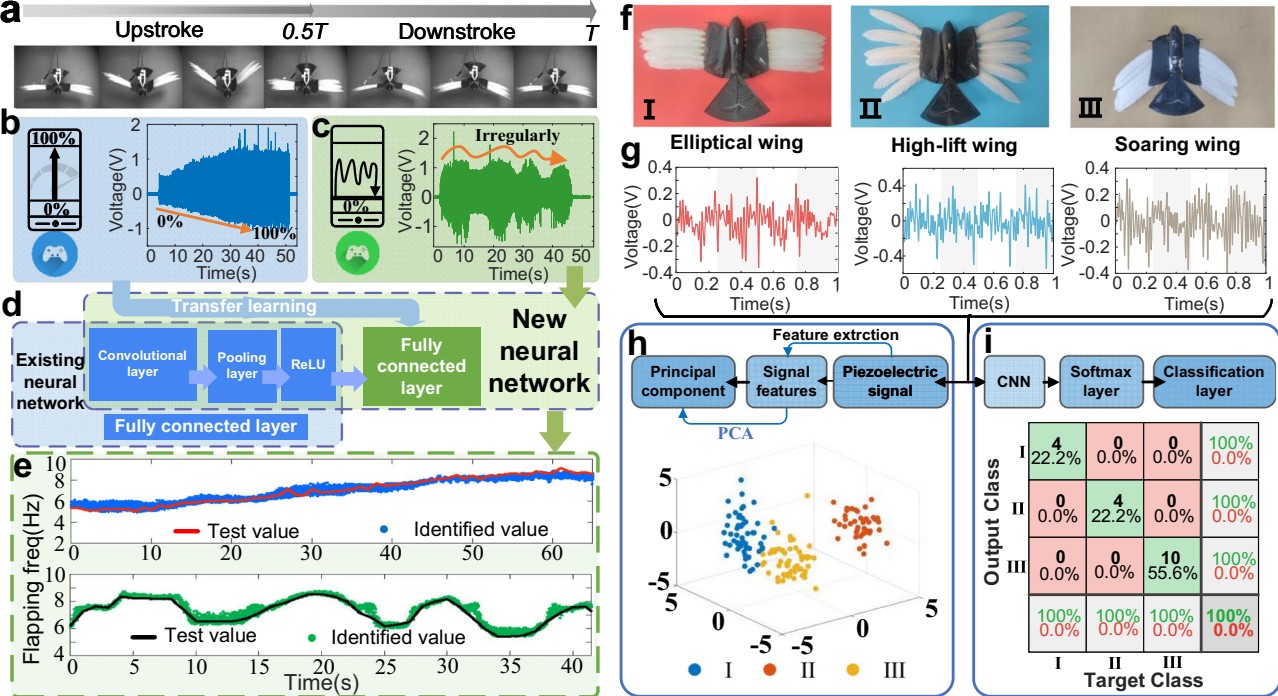

**Fig. 6 | Recognition of flapping frequency motion and wing morphology in the feathered flapping wing robot. a** Flapping period image of the robot with feather-PVDF biohybrid mechanoreceptor, *T* represents a complete flapping period. **b** Voltage signal of the robot undergoing approximately linear sweep frequency motion from 0% to 100% of power (blue). **c** Irregularly variable frequency motion voltage signal within the power range of the robot (green). **d** Schematic diagram of transfer learning and motion recognition process in the robot. The light blue blocks represent the network structure in Fig. 4, retaining all network layers except the fully connected layer and using the voltage signal from (**b**) of approximately linear sweep frequency motion from 0% to 100% power for weight training of the fully connected layer (transfer learning). A new network is obtained, and the voltage signal from irregularly variable frequency motion (**c**) is used to test the prediction performance of the transfer network. **e** Time-history curves of test and identified values for linear variable frequency motion (sliding step = 0.01 s, number of samples *n* = 6501) and irregular variable frequency motion (sliding step = 0.01 s, number of samples *n* = 4151). Source data are provided as a Source Data file. **f** Images of different wing morphologies, including elliptical wing (red, I), high-lift wing (blue, II), and soaring wing (brown, III), representing the flight states of birds in different environments. **g** Motion voltage signals of different wing morphologies. Source data are provided as a Source Data file. **h** Feature extraction and principal component analysis of the original data, resulting in a three-dimensional feature distribution map (number of samples *n* = 180). **i** Confusion matrix of wing morphology classification obtained through CNN network training.

biological hybrid perceptron. Given the diverse manifestations of tactile feedback, such as vibration, collision, touch, and pressure, these different forms provide rich avenues for the realization of homologous perception fusion. To explore the presentation of collision tactile feature differences in voltage signals, we collected corresponding voltage signals at a fixed flapping frequency (Fig. 6g), extracted 26 original features from the signals (Supplementary Table 5), reduced dimensionality to three dimensions using principal component analysis (Supplementary Fig. 32), and plotted the feature distribution of signals for the three wing morphologies in three-dimensional space (Fig. 6h). Noticeable differences in data features among the three wing morphologies ensure the feasibility of wing morphology classification recognition. Through CNN training on the collected signal samples and classification of the three wing morphologies, the confusion matrix (Fig. 6i) reveals a 100% recognition success rate for the three wing morphologies. This indicates that utilizing collision tactile perception can effectively achieve proprioceptive perception of wing morphologies in multi-feather wing structures, providing a method for motion perception in deformable wings.

The real-time embodied perception test of the feathered flapping robot was conducted in terms of flapping frequency, wind speed, pitch angle, and wing morphology (Fig. 7a and Supplementary Movies 7–10), with three classifications for each recognition category. The flapping robot, equipped with the feather-PVDF biohybrid perceptron, was fixedly mounted, and motion data was collected for 1 s using the data acquisition unit. The data were immediately fed into the pre-existing

CNN for recognition, and the recognition results were visualized in the interface (Fig. 7b–e). The recognition accuracies for the four key flight parameters (flapping frequency, wind speed, pitch angle, and wing morphology) were 100%. The specific recognition process can be found in the methods. Real-time recognition further illustrates that the biohybrid structure can achieve multisensory perception in the flapping robot. Simultaneously, the homologous perception of touch and proprioception provides a paradigm for the design of multisensory integration in robots.

To substantiate the viability of the bio-hybrid embodied perception approach in practical flight contexts, we meticulously crafted a feathered flapping-wing aircraft (Fig. 8a) capable of untethered flight (Fig. 8b, c and Supplementary Movie 11). This aircraft incorporates a gear-driven mechanism, a flight control board, piezoelectric signal acquisition, and high-impedance wireless transmission modules (Supplementary Figs. 33–35). With a total mass of 28.465 g (detailed mass distribution outlined in Supplementary Table 6), the PVDF film represents merely 0.79% of the total mass, whereas the data acquisition and wireless transmission module accounts for 5.15% of the total weight, with the bulk of the mass allocated to the airframe. To refine our mastery over the aircraft's lift and drag dynamics, pivotal for the subsequent development of a perception-motion closed-loop control system, we need to discern the relative flow velocities across the flapping wings, the aircraft's pitch angles, and the flapping frequencies. To calibrate the accuracy of embodied sensory identification, we devised a motion capture system ("Methods" and Supplementary

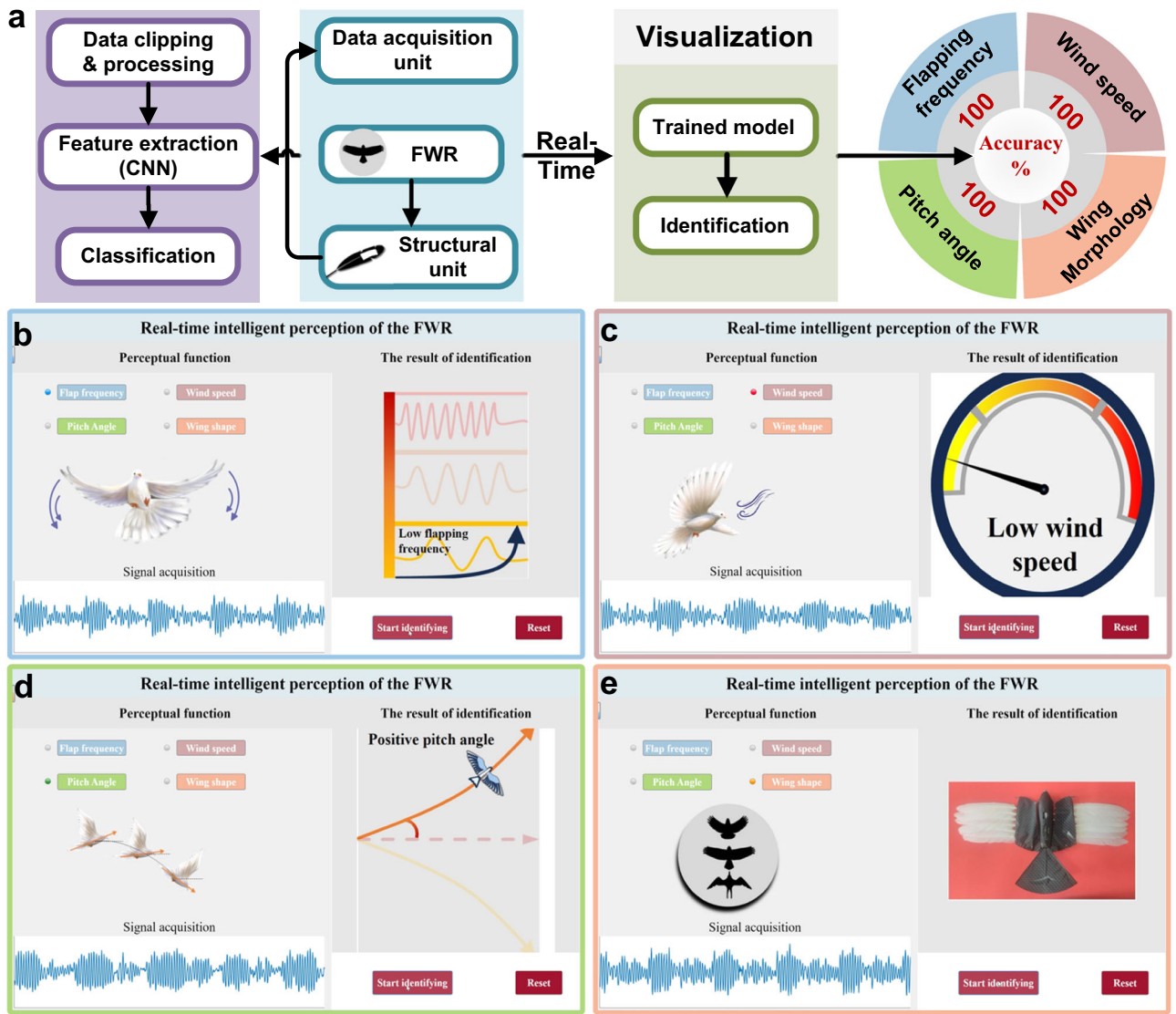

**Fig. 7 | Real-time recognition process, results, and visualization interface of feathered flapping wing robot. a** Real-time recognition process and recognition accuracy. **b** Visualization interface for real-time recognition of flapping frequency. **c** Visualization interface for real-time recognition of wind speed. **d** Visualization interface for real-time recognition of pitch angle. **e** Visualization interface for real-time recognition of wing morphology.

Fig. 36), utilizing an array of eight cameras to concurrently capture the aircraft's trajectory, track marker points, and reconstruct the three-dimensional motion coordinates, thereby computing the instantaneous flight speed and pitch angle for each frame. In indoor windless environments, we posit that the airflow velocity mirrors the aircraft's flying speed in magnitude but is opposite in direction. Direct surveillance of the instantaneous flapping frequency during flight presents challenges in flapping motion capture. However, we noted a high concordance between the flapping frequency gauged by an infrared laser displacement sensor and the predominant frequency of the piezoelectric signal when the aircraft engages in wing flapping without actual flight (Supplementary Fig. 37), yielding a relative error of 0.842%. Given that the flight behavior does not alter the correlation between the piezoelectric signal frequency and the actual flapping frequency, we equate the predominant frequency of the piezoelectric signal during flight to the flapping frequency measured by the displacement sensor.

Upon acquisition of the piezoelectric signals via wireless transmission and the corresponding indoor flight data from the motion capture system, these datasets were utilized to constitute a training ensemble (sliding window = 0.01 s). The convolutional neural network (CNN) architecture was congruent with the configuration delineated in Fig. 4g. The refined network model was then validated using indoor flight data independent of the training set. The comparison between motion-captured values and embodied perceived values is shown in Fig. 8d, with an average absolute error of 0.149 Hz for flapping frequency identification, 0.137 m s$^{-1}$ for velocity identification, and 2.360° for pitch angle identification. Our comparative analysis of flapping frequency and velocity demonstrates a high degree of concordance between motion captured and embodied perceived values, underscoring the robust accuracy of our methodologies. In contrast, the detection of pitch angle reveals a slightly elevated error, when compared to non-flight scenarios. We posit that this deviation may stem from the small fluctuations of pitch angle during flight, which complicates the discernment of minute airflow alterations resulting from small directional shifts. In addition, the concurrent presence of yaw, roll, and flapping motions in the aircraft's operational dynamics contributes to the nuanced challenges in assessing pitch angle through airflow direction and flow-induced vibrations. The intricate aerodynamic interactions and fluid-structure coupling unique to flapping-

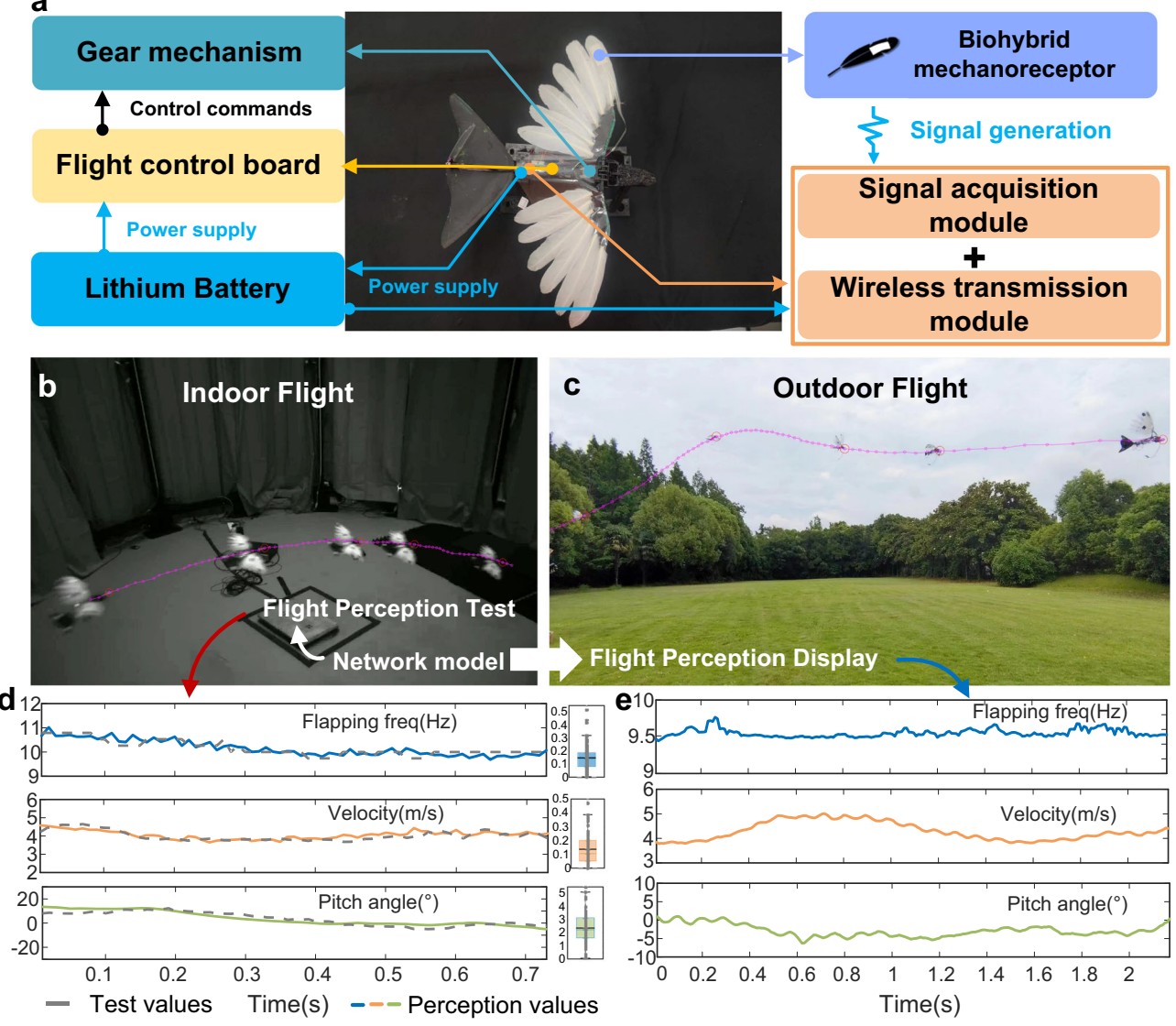

**Fig. 8 | Flight testing of the flapping-wing robot with feathered wings.**
**a** Structural component of the flapping-wing robot with feathered wings. **b** Indoor flight, the data independent of the training set is used as the indoor flight test data to obtain the corresponding network model. **c** Outdoor flight, using the verified network model in Fig. (**b**) for identification and display. **d** Comparison of the test and perception values in Fig. (**b**), number of samples $n = 73$, the identification error distributions of flapping frequency (blue), velocity (orange), and pitch angle (green) are shown on the right side, the box plots show the maximum, minimum, 25th percentile, 75th percentile, and median error values for each category, the black horizontal line represents the perception average error value. Source data are provided as a Source Data file. **e** The perception results of outdoor flight data in Fig. (**c**) are displayed, number of samples $n = 217$. Source data are provided as a Source Data file.

wing flight, relative to their fixed-wing counterparts, indeed pose a more complex identification challenge. We also conducted outdoor flight tests, applying the CNN model established from indoor flights to identify outdoor flight parameters (Fig. 8e). The practical flight tests of the bio-hybrid flapping-wing robot demonstrate the efficacy of the feather-PVDF embodied perception method, which will aid in the flight state monitoring and reflex control of miniature flapping-wing aircraft, and provide broader insights into the development of embodied perception in robotics.

## Discussion

Natural feathers, shaped and evolved over one hundred million years, profoundly influence avian flight. The agile, flexible, and efficient flight behaviors, as well as the perceptual methods of birds, have inspired the development of drones. This research introduces a biohybrid multisensory fusion strategy, uniting natural feathers with flexible

piezoelectric films to enable embodied perception in feathered flapping-wing robots. The employed feather-PVDF biohybrid mechanoreceptor closely mirrors avian traits, aiming to maintain natural aerodynamic performance and structural functionality. The study delves into heterointerface connection properties between feathers and PVDF, highlighting excellent peeling strength, durability, and electromechanical performance. The responsive behavior of feather structures to airflow and flapping stimuli translates into distinctive voltage signals on flexible piezoelectric film. Vibration responses of feather branch structures and inter-feather collisions amplify signal differences under diverse stimuli, forming the basis for signal recognition across varied tactile environments and body movements. Regression recognition of voltage signals under different flapping frequencies, headwind speeds, and angles is achieved through CNN and GWO algorithms. The trained network adeptly recognizes temporally varying motion patterns. The biohybrid

embodied perception achieves high-precision recognition of the flight state for a flapping-wing robot at a low weight cost. The multisensory fusion perception based on the homology of touch and kinesthesia further enhances the multifunctionality of structural units. Through untethered flight tests, we validated the feasibility of bio-hybrid perception recognition.

Biohybrid perception can also be applied to monitor real bird behaviors. Currently, our understanding of bird flight principles is not complete, and direct observation or domestication is time-consuming. Piezoelectric materials have been proven to be biocompatible[63]. The use of biohybrid perception and wireless transmission technology not only offers new methods for bird monitoring and biological research but also aligns with the principles of biomimetic robotic design. Bio-mimetic robotic design draws inspiration from natural organisms and biological materials, celebrated for their intricately interconnected systems capable of diverse functionalities[64]. The integration of intelligent and biomimetic materials offers insights into the embodied intelligence design of flapping-wing robots. Our feather-PVDF mechanoreceptor is particularly well-suited for discrete morphing wings with elastic tendons and frictional hooking, fostering potential synergistic advancements in feathered flapping-wing robotics[5,11]. With the continuous advancement of artificial intelligence models, multi-modal embodied intelligence emerges as an effective design paradigm in the realm of robotics[65]. This approach underscores the perception-action loop and introduces a multi-level evolution design framework for materials and machines[66]. By integrating interactions between the body and the environment to obtain sensory feedback, thereby serving the control system, an effective neural control strategy is established. However, the practical integration of these biohybrid perception functions into reflex-based control will necessitate extensive flight testing[11,12,67] and the development of lightweight data collection, transmission systems, and responsive multifunctional actuators. Furthermore, achieving autonomous interaction between FWRs and the environment, fostering adaptive intelligent behavior[68], will require more efficient and stable intelligent algorithms. This trend will further propel the integration of multifunctionality and intelligence in future autonomous robotic systems, encompassing aspects of sensing, decision-making, response, and energy.

## Methods

### Microscale characterization
We conducted microscale characterization using an optical microscope (GP-304K, Kunshan Gao pin Precision Instrument Co., Ltd.). PVDF films were affixed to feathers, and in the regions where the piezoelectric films were attached, the feathers were cut along both parallel and perpendicular directions to the feather axis. Cross-sectional microscopic views were captured in both directions. Slow peeling was performed on the observation platform, capturing microscopic views of the breaking filaments during the peeling process. Complete feathers were cut, and after surface gold coating, the microstructures on the feather surface were observed using a low-vacuum, high-resolution field-emission scanning electron microscope (NOVA Nano SEM 230, USA).

### Peeling experiment
Preparation of carbon fiber sheets, PET, and feathers involved cutting each material into 2.5 cm × 6 cm dimensions based on the dimensions of the piezoelectric film and peeling fixture. The peeling experiment was conducted using a tensile machine (HengYi-HY0580). The peeling fixture was secured beneath the tensile machine, and after closely adhering each material to the PVDF, a portion of the peeling sample, excluding PVDF, was fixed in the fixture. The tensile end of the machine was fitted with a clamp, and a force sensor (Transcell Technology, Inc.) with a range of 20 N was used. The PVDF lead was bent upward, adjusted to a vertical position, and clamped with the clamp, ensuring

secure installation. The PVDF was then peeled off at a constant rate of 180° and 50 mm/min. The peeling force-displacement curve was observed, and when the PVDF detached from the sample, the curve returned to zero, and the upward movement was halted. After replacing the peeling sample, the clamp end was retracted to the initial position, and the process was repeated. Five peeling experiments were conducted for each material, and the average peeling force-displacement curve was plotted for each material based on the averages of the five experiments.

### Fatigue experiment
Utilizing 3D printing, two clamping plates (PLA, 1.5 cm × 4 cm) were fabricated to secure the base of the feather. These plates were firmly fastened to the shaker table using screws. The PVDF piezoelectric film was affixed to the feather, and its leads were connected to the data acquisition system channels. A mass block weighing 0.3 g was fixed at the leading edge of the feather to increase its bending angle and accelerate fatigue. The experimental setup involved sequentially connecting a signal generator (RIGOL DG-1032), a power amplifier (SINOCERA YE55874A), and a shaker table (SINOCERA JZK-50). The signal generator frequency was set to 30 Hz. The voltage knob of the power amplifier was adjusted until the shaker table operated stably, and the output voltage signal was then collected.

### Bending experiment
A single feather was secured beneath the tensile testing machine, and a PVDF piezoelectric film was affixed to the feather. The distance from the front end of the PVDF film to the fixed end of the feather was considered the flexural force arm. The feather was compressed downward at a rate of 50 mm/min until the feather axis yielded. The experiment was repeated with the reverse side of the feather, maintaining the same conditions. For the multi-feather wing structure, multiple feathers were fixed beneath the tensile testing machine, and PVDF piezoelectric films were attached to the feathers at the same position, ensuring consistent flexural force arms. The flexural experiment was conducted at the same rate. In addition, experiments were performed without PVDF attachment, both for individual feathers and the multi-feather wing structure, ensuring identical force arm lengths and compression rates.

### Self-powered experiment
The wing was installed onto the actuation mechanism, and the PVDF piezoelectric film was affixed to the specified feather. The actuation mechanism, along with the support platform, was securely fixed on the optical platform. Considering that the output of the piezoelectric unit is alternating current, we converted the AC voltage output of the piezoelectric film into direct current using a rectifier bridge and connected it to a capacitor to store energy. We constructed the rectifier bridge circuit on a breadboard, as shown in Fig.4b, with the input side connected to the leads of the piezoelectric film. After connecting the power supply, we adjusted the speed controller knob to achieve a high flapping frequency. Charging was halted after a certain duration, and the capacitor's output terminal was connected to the electronic devices to be powered. The driven temperature and humidity digital sensor and the digital watch had rated voltages of 1.5 V.

### Flapping signal acquisition
Powered by a 12 V DC power supply, the motor speed is controlled using a digital speed controller. One end of the digital speed controller is connected to the 12 V power supply, while the other end is connected to the motor drive mechanism pins. Upon powering, the speed can be controlled using the digital speed controller's knob, with readings ranging from 0 to 100. The flapping stroke is monitored using a laser displacement sensor (KEYENCE IL-300). The infrared laser emitter is fixed above the drive mechanism, allowing the infrared laser

to shine downward onto the reflective tape. The sensor amplifier is connected to the infrared laser emitter, and when there is a vertical change in position between the reflective tape and the infrared laser emitter, the sensor amplifier outputs a corresponding voltage change. The PVDF piezoelectric film is attached to the specified feather, and the leads from the piezoelectric film and the output lead from the sensor amplifier are simultaneously connected to the data acquisition system (East China DH5922D) channels. The voltage signals generated by the piezoelectric film and the laser displacement sensor are collected separately. The adjustment of the pitch angle is achieved by using a tilting platform. The support platform with the drive mechanism is fixed on the tilting platform, and the pitch angle of the flapping mechanism can be changed by adjusting the tilting platform's knob. This entire setup is placed in the wind tunnel test section, with a total length of 4.9 m. The wind speed is regulated by a frequency converter in the wind tunnel, and the actual wind speed is measured by a split-type anemometer (JAHAN W410D).

### Time-varying frequency flapping experiment

We utilized an Arduino development board and an L298N module to achieve the precise driving and control of the flapping motor. The Arduino was directly powered by the computer with 5 V, and considering voltage drop, one end of the L298N module was powered by a 14 V power supply, while the other end was connected to the positive and negative poles of the flapping motor. Code was written in the Arduino IDE to control the variation pattern of the motor's positive and negative voltage inputs, thereby regulating the motor's variable-speed movement. The PVDF piezoelectric film was affixed to the designated feather, and IL series laser displacement sensors were employed to record the flapping stroke. The leads were connected to the data acquisition system channel, and the computer simultaneously collected the voltage signals generated by the piezoelectric film and laser displacement sensors. On the computer side, the control imported the Arduino IDE code file, signal acquisition commenced, the flapping mechanism initiated movement continued for several complete cycles, and data acquisition concluded, saving the data for analysis.

### Feathered flapping-wing robot experiment

The feathered flapping wing robot has a total weight of 14.5 g, with a flapping amplitude of 55°, equipped with a 1.6-watt micro-coreless motor and an aluminum heat sink. The motor, under no-load conditions, can achieve a speed of up to 53,000 rpm. The connecting structure between the body and feathers is made of carbon fiber material using 3D printing. One end of this structure connects to the body slot, while the other end has six 4 mm diameter holes for feather insertion, with different wing morphologies having distinct hole distribution. Once the wing structure is stably connected to the body, the power switch on the body is turned on. By connecting a controller via Bluetooth on a mobile phone, the flapping frequency of the feathered flapping wing robot can be adjusted within the range of 0%–100%. A metal rod is fixed at one end and connected to the body through a clamp at the other end. High-speed cameras (Qian Yan Lang M220 model by Hefei General Vision Technology Co., Ltd.) and a high-speed data acquisition control system (Revealer Control Center) are used to capture the flapping cycle. PVDF piezoelectric films are attached to specific feathers, and their leads are connected to the data acquisition system channel. The computer continuously collects the voltage signals generated by the piezoelectric film.

### Real-time recognition experiment

Fix the feathered flapping wing robot with the feather-PVDF biohybrid perceptron. Connect one end of the data acquisition unit (USB2AD7606, Shenzhen Anybo Electronic Technology Co., Ltd.) to the PVDF lead and the other end to the computer. This data acquisition board uses the AD7606 module to achieve high-sampling-rate conversion from analog to digital signals. USB serial communication is implemented using the STM32 chip. The data acquisition unit starts collecting when the light begins to flash and stops when the light stops flashing. Utilizing MATLAB to create a GUI interface, which includes four recognition options: flapping frequency, wind speed, pitch angle, and wing morphology. Adjust the flapping frequency using the mobile controller, set three power levels at 10, 40, and 70%, click the flapping frequency button on the GUI interface, enter the flapping frequency recognition module, click the start recognition button, and the data acquisition unit performs one-second data collection. It immediately returns the data to the computer, classifies and recognizes it using the existing neural network, and finally provides feedback on the GUI interface with the corresponding result (low flapping frequency, medium flapping frequency, high flapping frequency) to complete the flapping frequency recognition. Use a three-level adjustable fan to control the environmental wind speed, and recognize the wind speed under the condition of 10% flapping power (low wind speed, medium wind speed, high wind speed); adjust the headwind angle of the flapping wing robot and recognize the pitch angle in the low wind speed setting (positive pitch angle, zero-degree pitch angle, negative pitch angle); adjust the wing morphology and recognize the wing morphology under the condition of 10% flapping power (elliptical wing, high-lift wing, and soaring wing).

### Indoor and outdoor flight test

Indoor flight tests were conducted within a camera cage array, which is a hexadecagon framework with a side length of 0.98 m, an outer circle diameter of 5.12 m, and a height of 3.52 m. The array was equipped with eight MER-160-227U3M monochrome cameras, each with a resolution of 1440 × 1080p, featuring Sony IMX273 CMOS sensors with an effective area of 1/1.29 inches. The cameras used Myutron FV0420 lenses with a focal length of 4.16 mm and a maximum aperture of F/2.0. These eight cameras were distributed in a crisscross pattern, with four positioned at a higher level (1.95 m) and four at a lower level (0.40 m), operating at a frame rate of 100 Hz. The cameras were synchronized with the computer and the flight signal acquisition/wireless transmission module. The aircraft was controlled to pass through the array while recording multi-angle video and sensor data of the flight process, detailed processing can be seen in Supplementary Information. Outdoor flight tests were conducted on a spacious lawn under windless conditions. The aircraft was flown for a certain distance, with the flight video recorded using a smartphone camera and flight sensor data collected on a computer. The neural network structure is consistent with Fig. 4g, the training set is derived from the data collected from indoor flight, and the identification performance of the three networks is tested using the complete indoor flight data independent of the training data set.

## Data availability

All the data supporting the findings of this study are available within the main text and its Supplementary Material. Source data are provided in this paper.

## Code availability

The codes for data preprocessing, training, and testing CNN network models and the dataset have been deposited in the Zenodo repository: https://doi.org/10.5281/zenodo.13883469.

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

## Acknowledgements

We sincerely appreciate the National Key R&D Program of China (Grant No. 2023YFF0616800, Z.Y.) and the Oceanic Interdisciplinary Program of Shanghai jiao Tong University (Project No. SL2022MS001, Z.Y.) for the financial support to this study.

## Author contributions

Q.L., T.T., and Z.Y. conceived the ideas and designed the research. Q.L. and T.T. established the theoretical model. Q.L. performed the experiments. Q.L., T.T., and Z.Y. analyzed the data. Q.L., T.T., B.W., and Z.Y. interpreted the results and wrote the manuscript with input from all authors. Z.Y. and T.T. supervised the study.

## Competing interests

The authors declare no competing interests.
