## [Transparent Peer Review file · Nature Communications]

Avian-Inspired Embodied Perception in Biohybrid Flapping-Wing Robotics

Corresponding Author: Professor Zhimiao Yan

Version 0:

Reviewer comments:

Reviewer #1

(Remarks to the Author)

This manuscript offers a promising vision of the integration of embodied perception within biohybrid flapping wing robotics. However, the approaches and results presented—ranging from sensor design (utilizing a simplistic parallel capacitance sensor) to its integration (relying on wired connections and cumbersome DAQ systems) and the adopted signal processing methodology (basic Convolutional Neural Networks)—fall short of the expectations for a publication of this prestigious journal. Furthermore, the rationale behind the selection of a biohybrid flapping-wing framework remains unclear to the reviewer. The application of such soft sensors to actual feathers could potentially interfere with the flight dynamics of living birds, causing undue harm. Conversely, for biomimetic avian robots, emulating feather structures in such detail seems unnecessary.

Other specific comments:

1. The detailed sensor structure and sensing principle of the PVDF is not clear. There seem to be two parallel silver electrodes from Fig. 2. Is this based on parallel capacitive sensing? If this is the case, the novelty of this key aspect is insufficient as such kinds of sensors have been very commonly reported.
2. Fig. 4 i, There seem to be considerable errors, especially for wind speed and pitch angle estimation. For some results, overlapping between levels is evident. There should be criteria regarding the usability of such results.
3. Line 81, what is PVDF?
4. The detailed architecture of the network is not clear.
5. Fig.2 j, it is not visible regarding the detailed waveforms of the cycling test. Is that a linear response or a nonlinear one?
6. Most of the figures are of low quality/resolution, hindering information extraction.

Reviewer #2

(Remarks to the Author)

Interesting work, but there is a mismatch between the title and the content of the article. Specifically the title refers to "flapping-wing robotics" while the body of the text only shows flapping devices used to test the hybrid systems without any demonstration of flight. As written, the article is not suitable for publication in Nature Communications, as it demonstrates development of a sensor and some data processing methods.

For the work to be suitable, it would need to grow in one of two possible directions:

- demonstrate the device in an actual flapping wing robot capable of lift, even if tethered to a power source. The data acquired during this type of demonstration would be a significant proof of the sensing method and its value to the broader community.
- drop the focus on robotics, and broaden the bio-inspired perspective. One suggestion is to look at the mechanosensory capability of fishes using their lateral line system, one good reference to start with is Webb, J.F., Maruska, K.P., Butler, J.M. and Schwalbe, M.A., 2021. The mechanosensory lateral line system of cichlid fishes: from anatomy to behavior. The behavior, ecology and evolution of cichlid fishes, pp.401-442. The parallels between lateral line and feather sensing in moving fluids would be a great source of inspiration across disciplines, from biology to engineering.

General suggestion to improve the figures:

- the text in Fig. 4, 5, 6 and 7 is difficult to read, the resolution of the figures is poor.

- the difference between c and d in Figure 2 is not immediately apparent, could be clarified.
- the purpose of figures h and i in the same figure is unclear, because it doesn't compare systems with and without PVDF
- adding sensing capability to feathers is one thing, however the major advantage in natural systems is the ability to rapidly change between wing shapes, as shown in Figure 6 f. Does the specific PVDF method lead the authors to believe one specific morphing wing technology is more suitable for feathers embedded with this type of sensing? What downstream design impact does adding the PVDF and learning algorithm have on robot design?

Reviewer #3

(Remarks to the Author)

The manuscript reports on experiments with a flapping wing robot with piezoelectric PVDF films integrated into its feathered wings. The authors report on the manufacturing processes of the wings and especially the film bonding procedure. Experiments at various flapping frequencies, pitch angles, wind velocities and feather arrangements were conducted using the tethered robot. Employing Convolutional Neural Networks (CNN) aided by the Grey Wolf optimization algorithm for training, the authors show that a one-dimensional CNN can predict all the studied states (flapping frequency, pitch angle, wind velocity, feather arrangement) with relatively small errors from the real states. Further, the authors show the prediction gives accurate prediction of the states even when done in (near) real time (1s measurement and then immediate evaluation). Finally, the authors demonstrate that the piezoelectric sensors can even generate some power, that can be reused e.g. by powering other on board devices such as other sensors.

In the reviewer's opinion, the work represents a significant step forward in embodied perception and may lead to lighter and smaller flapping-wing robots in the future. Such robots will have limited lifting capabilities and so an embodied sensor that could sense multiple states is highly desirable.

The authors conducted the study systematically and thoroughly, the methodology, data analysis and its interpretation is sound, and overall supports their conclusions and claims (with minor exceptions, which I list further). Enough detail is provided for the work to be reproduced (again, with minor exceptions listed below). I recommend the work for publication once my minor comments below are addressed in the final revised manuscript.

Matej Karasek, Flapper Drones

Minor comments:

- The claims made about the real-time estimation suggest the CNN can estimate continuous states (e.g. abstract, line 25). However, the results presented show that the data is segmented first (1 s) and then evaluated, so the output will be delayed and downsampled. For feedback control, such delay could cause instability, could the authors discuss why 1s segments were chosen? Could the authors estimate what would be the minimal segment duration (e.g. in function of the flapping frequency) for this method to work reliably?
- The authors speak about low weight of their solution (abstract, line 26, and lines 462-3), but I did not manage to find any mass specification of the added sensors. E.g. a mass breakdown table or percentage of the sensor mass to the total wing- and robot mass should be presented. In a free flying robot, such added mass would encompass also all the required circuitry to digitize and process the sensor signals and estimate the states.
- The authors emphasize the ability of the piezoelectric sensors to recover energy. To quantify such benefit, could the authors estimate how much power could be recovered, and how it compares e.g. to the power needed to flap the wings? (section at line 530)
- It is not entirely clear to the reviewer, whether a single trained CNN can estimate all the studied states (which the reviewer assumes), or whether for each case a separate training was used. Please make this more explicit in the text.
- line. 467 the implications to environmental monitoring are not clear to the reviewer, please explain in more detail.
- Feathered robot design (section at line 573): is this own robot design of the authors, or an adapted off-the-shelf robot? In both cases, please refer to more details about the robot, or report the robot make and model. It seems to be a different robot than the one used in the wind tunnel experiment (Figure 3c)
- Can you give more details about the wind tunnel type, and possibly in which facility it is located?

Version 1:

Reviewer comments:

Reviewer #1

(Remarks to the Author)

I appreciate the effort the authors have made to reply my previous comments. I feel that the quality of the paper has been improved to some extent after revision. However, the concern is the major issues raised in the previous version have not

been fully addressed and the reviewer is still not convinced about the novelty of the sensor and signal processing aspect. This seems to be a good application, but only a good application and a stack of trivial work and analysis cannot elevate the key novelty of the work.

Reviewer #2

(Remarks to the Author)

I appreciate the authors taking the time to re-develop hardware to respond in earnest to all of the reviewer feedback. The new direction, supported by the development of novel hardware, on which their innovation can be tested, is a significant achievement in the field. As a result, the work is suitable for publication in this journal. I see no other errors that need correcting, and the new message for the work is significantly more compelling - the work is ready for publication without any additional changes.

Reviewer #3

(Remarks to the Author)

The authors have answered all my comments to my full satisfaction and improved the manuscript accordingly; the limitations of the methodology and applicability of the results to FWMAVs got further clarified. The newly added flight experiments and other details added in response to the other reviewers' comments, further strengthen the scientific contribution of the manuscript.

I recommend the paper for publication.

I also recommend publishing all the raw experimental data as a dataset, such that interested readers can duplicate the results following the published methodology.

Reviewer #1 (Remarks to the Author):

This manuscript offers a promising vision of the integration of embodied perception within biohybrid flapping wing robotics. However, the approaches and results presented—ranging from sensor design (utilizing a simplistic parallel capacitance sensor) to its integration (relying on wired connections and cumbersome DAQ systems) and the adopted signal processing methodology (basic Convolutional Neural Networks)—fall short of the expectations for a publication of this prestigious journal. Furthermore, the rationale behind the selection of a biohybrid flapping-wing framework remains unclear to the reviewer. The application of such soft sensors to actual feathers could potentially interfere with the flight dynamics of living birds, causing undue harm. Conversely, for biomimetic avian robots, emulating feather structures in such detail seems unnecessary.

Other specific comments:

1. The detailed sensor structure and sensing principle of the PVDF is not clear. There seem to be two parallel silver electrodes from Fig. 2. Is this based on parallel capacitive sensing? If this is the case, the novelty of this key aspect is insufficient as such kinds of sensors have been very commonly reported.
2. Fig. 4 i, There seem to be considerable errors, especially for wind speed and pitch angle estimation. For some results, overlapping between levels is evident. There should be criteria regarding the usability of such results.
3. Line 81, what is PVDF?
4. The detailed architecture of the network is not clear.
5. Fig.2 j, it is not visible regarding the detailed waveforms of the cycling test. Is that a linear response or a nonlinear one?
6. Most of the figures are of low quality/resolution, hindering information extraction.

Reviewer #1 (Remarks on code availability):

I don't have access to any codes.

Reviewer #2 (Remarks to the Author):

Interesting work, but there is a mismatch between the title and the content of the article. Specifically the title refers to "flapping-wing robotics" while the body of the text only shows flapping devices used to test the hybrid systems without any demonstration of flight. As written, the article is not suitable for publication in Nature Communications, as it demonstrates development of a sensor and some data processing methods.

For the work to be suitable, it would need to grow in one of two possible directions:

- demonstrate the device in an actual flapping wing robot capable of lift, even if tethered to a power source. The data acquired during this type of demonstration would be a significant proof of the sensing method and its value to the broader community.

- drop the focus on robotics, and broaden the bio-inspired perspective. One suggestion is to look at the mechanosensory capability of fishes using their lateral line system, one good reference to start with is Webb, J.F., Maruska, K.P., Butler, J.M. and Schwalbe, M.A., 2021. The mechanosensory lateral line system of cichlid fishes: from anatomy to behavior. The behavior, ecology and evolution of cichlid fishes, pp.401-442. The parallels between lateral line and feather sensing in moving fluids would be a great source of inspiration across disciplines, from biology to engineering.

General suggestion to improve the figures:

- the text in Fig. 4, 5, 6 and 7 is difficult to read, the resolution of the figures is poor.

- the difference between c and d in Figure 2 is not immediately apparent, could be clarified.

- the purpose of figures h and i in the same figure is unclear, because it doesn't compare systems with and without PVDF

- adding sensing capability to feathers is one thing, however the major advantage in natural systems is the ability to rapidly change between wing shapes, as shown in Figure 6 f. Does the specific PVDF method lead the authors to believe one specific morphing wing technology is more suitable for feathers embedded with this type of sensing? What downstream design impact does adding the PVDF and learning algorithm have on robot design?

Reviewer #3 (Remarks to the Author):

The manuscript reports on experiments with a flapping wing robot with piezoelectric PVDF films integrated into its feathered wings. The authors report on the manufacturing processes of the wings and especially the film bonding procedure. Experiments at various flapping frequencies, pitch angles, wind velocities and feather arrangements were conducted using the tethered robot. Employing Convolutional Neural Networks (CNN) aided by the Grey Wolf optimization algorithm for training, the authors show that a one-dimensional CNN can predict all the studied states (flapping frequency, pitch angle, wind velocity, feather arrangement) with relatively small errors from the real states. Further, the authors show the prediction gives accurate prediction of the states even when done in (near) real time (1s measurement and then immediate evaluation). Finally, the authors demonstrate that the piezoelectric sensors can even generate some power, that can be reused e.g. by powering other on board devices such as other sensors.

In the reviewer's opinion, the work represents a significant step forward in embodied perception and may lead to lighter and smaller flapping-wing robots in the future. Such robots will have limited lifting capabilities and so an embodied sensor that could sense multiple states is highly desirable.

The authors conducted the study systematically and thoroughly, the methodology, data analysis and its interpretation is sound, and overall supports their conclusions and claims (with minor exceptions, which I list further). Enough detail is provided for the work to be reproduced (again, with minor exceptions listed below). I recommend the work for publication once my minor comments below are addressed in the final revised manuscript.

Matej Karasek, Flapper Drones

Minor comments:

- The claims made about the real-time estimation suggest the CNN can estimate continuous states (e.g. abstract, line 25). However, the results presented show that the data is segmented first (1 s) and then evaluated, so the output will be delayed and downsampled. For feedback control, such delay could cause instability, could the authors discuss why 1s segments were chosen? Could the authors estimate what would be the minimal segment duration (e.g. in function of the flapping frequency) for this method to work reliably?

- The authors speak about low weight of their solution (abstract, line 26, and lines 462-3), but I did not manage to find any mass specification of the added sensors. E.g. a mass breakdown table or percentage of the sensor mass to the total wing- and robot mass should be presented. In a free flying robot, such added mass would encompass also all the required circuitry to digitize and process the sensor signals and estimate the states.

- The authors emphasize the ability of the piezoelectric sensors to recover energy. To quantify such benefit, could the authors estimate how much power could be recovered, and how it compares e.g. to the power needed to flap the wings? (section at line 530)

- It is not entirely clear to the reviewer, whether a single trained CNN can estimate all the studied states (which the reviewer assumes), or whether for each case a separate training was used. Please make this more explicit in the text.

- line. 467 the implications to environmental monitoring are not clear to the reviewer, please explain in more detail.

- Feathered robot design (section at line 573): is this own robot design of the authors, or an adapted off-the-shelve robot? In both cases, please refer to more details about the robot, or report the robot make and model. It seems to be a different robot than the one used in the wind tunnel experiment (Figure 3c)

- Can you give more details about the wind tunnel type, and possibly in which facility it is located?

RESPONSE TO REVIEWER COMMENTS

Reviewer #1

This manuscript offers a promising vision of the integration of embodied perception within biohybrid flapping wing robotics. However, the approaches and results presented—ranging from sensor design (utilizing a simplistic parallel capacitance sensor) to its integration (relying on wired connections and cumbersome DAQ systems) and the adopted signal processing methodology (basic Convolutional Neural Networks)—fall short of the expectations for a publication of this prestigious journal. Furthermore, the rationale behind the selection of a biohybrid flapping-wing framework remains unclear to the reviewer. The application of such soft sensors to actual feathers could potentially interfere with the flight dynamics of living birds, causing undue harm. Conversely, for biomimetic avian robots, emulating feather structures in such detail seems unnecessary.

Response: We sincerely appreciate your positive feedback and valuable suggestions regarding our research. In response to the issues raised on scientific advance, the scientific mechanism of the bio-hybrid perception method proposed in this paper includes two parts: the feather-based vibration structure for transmitting deformation signals and the positive piezoelectric effect for electromechanical conversion. Feathers manifest pronounced vibration patterns and frequencies when subjected to airflow, with adjacent vibrating feathers acting as coupled oscillators that undergo collisions and nonlinear interactions [1], thereby augmenting the differentiation of mechanoreception. It is due to the microscopic barbule structure of feathers that an excellent adhesive mechanical performance is achieved at the heterogeneous interface between the feather surface and the glue filament layer of the piezoelectric film, ensuring that the piezoelectric signal remains unaltered even after 10,000 cycles. Consequently, we harness feathers as intricate structures for the sensing of airflow and vibration, and propose the feather-PVDF bio-hybrid embodied perception method with significant biomechanical value for flapping wing aircraft.

In response to your concern regarding wire connections and data acquisition integration, we have developed a flapping-wing robot to transition from a wired platform to untethered real flight in the revised manuscript. The overall weight of the robot is 28.465 g, with a signal acquisition and wireless transmission module of 1.465 g. We conducted indoor and outdoor

flight experiments, which validated the feasibility of embodied perception during actual flight. In the realm of algorithms, we adopt Convolutional Neural Networks (CNNs) instead of the prevalent feedforward neural networks such as DNN and MLP, as seen in recent studies on flexible sensor-based flight state recognition [2,3]. CNNs benefit from local connections and parameter sharing, thereby reducing computational complexity and the risk of overfitting. To optimize neural network hyperparameters, we implement the Grey Wolf Optimizer, which efficiently navigates the search space. This combination yields satisfactory recognition outcomes.

The selection of the bio-hybrid flapping wing framework is based on several key considerations. Firstly, the rationale for introducing the feather structure as the fundamental vibrating structure of the sensor has been explained in the first paragraph of our response. Additionally, PVDF piezoelectric films, known for their lightweight, flexibility, high sensitivity, and stability, are widely used in tactile sensing, biomedical sensing, and wearable flexible electronics, and have been proven to be biocompatible [4,5]. In the context of aircraft technology, the frictional interlocking between feathers can enable smooth and stable deformations [6], while their asymmetrical structure provides superior aerodynamic load-bearing capacity. Thus, feather structures are conveniently adaptable to discrete-unit based continuous morphing wing technology for agile maneuverability [7]. Moreover, the utilization of feather materials in the construction of flapping wing aircraft wings aids in achieving a high level of mimicry, aiming for indistinguishability within bird flocks. The bio-hybrid framework integrates multidisciplinary insights from biology, ecology, biomechanics, and robotics, thereby facilitating the advancement of highly biomimetic and stealthy morphing flapping wing aircraft systems.

[1] Clark, C. J., Elias, D. O. & Prum, R. O. Aeroelastic flutter produces hummingbird feather songs. *Science*. **333**, 1430-1433 (2011).

[2] Gong, Z. et al. Flexible calorimetric flow sensor with unprecedented sensitivity and directional resolution for multiple flight parameter detection. *Nat. Commun.* **15**, 3091 (2024).

[3] Xiong, W. et al. Bio-inspired, intelligent flexible sensing skin for multifunctional flying perception. *Nano Energy*. **90**, 106550 (2021).

[4] Mohammadpourfazeli, S. et al. Future prospects and recent developments of polyvinylidene fluoride (PVDF) piezoelectric polymer ; fabrication methods , structure , and electro-mechanical properties. *Rsc*

Adv. **13**, 370-387 (2022).

[5] Yang, T. et al. Hierarchically structured PVDF / ZnO core-shell nanofibers for self-powered physiological monitoring electronics. *Nano Energy*. **72**, 104706 (2020).

[6] Matloff, L. Y. et al. How flight feathers stick together to form a continuous morphing wing. *Science*. **367**, 293 (2020).

[7] Chang, E., Matloff, L. Y., Stowers, A. K. & Lentink, D. Soft biohybrid morphing wings with feathers underactuated by wrist and finger motion. *Sci. Robot.* **5**, 1246(2020).

*We added the description of bio-hybrid conception in **Introduction** as follows:*

(page 2, lines 41-46) “Using discrete feathers addresses issues like wrinkling in membrane wings during contraction. Additionally, these adaptable feathers significantly enhance aircraft maneuverability and agility, allowing faster and tighter turns⁹ and enhancing flight concealment. These studies not only provide fresh insights for the advancement of future unmanned aerial vehicles but also deepen our understanding of the fundamental principles guiding avian flight¹⁵.”

(pages 2-3, lines 68-85) “Research indicates that most feathers exhibit distinct vibration modes and frequencies under varying flow velocities and directions. The adjacent vibratory feathers might operate as coupled oscillators, culminating in collisions that engender nonlinear interactions⁴². Feathers exhibit remarkable responsiveness to external airflow while boasting lightweight, tear-resistant characteristics^{43,44}. Moreover, they possess a restitution property, enabling their surface morphology to be restored following disorder, merely through the application of gentle stroking or tidying. Drawing upon natural feather biomaterials and structures to enhance the differentiation of mechanical perception, integrating flexible lightweight self-powered PVDF to emulate the functions of avian wing mechanoreceptors, this study proposes a biohybrid embodied perception approach with deep learning (Fig. 1a and Supplementary Movie 1). The investigation focuses on the heterogeneous interface connection properties of the feather-PVDF biohybrid mechanoreceptor, encompassing characteristics like peeling strength, fatigue durability, and electromechanical conversion efficiency. Based on avian-inspired FWRs with feathered wings, we achieve a biohybrid tactile and kinesthetic system with real-time classification and visualization of flapping frequency, wind speed, pitch angle, and wing shape (Fig. 1b). Additionally, we complete accurate perception of flapping frequency and relative flow velocity during untethered flight. This biohybrid embodied perception design holds potential for advancing the development of more lightweight, integrated, stealthy, and dexterous biomimetic flapping-wing air vehicles.”

*We added the discussion of algorithm aspect in **Main text** as follows:*

(page 11, lines 304-313) “Compared to fully connected feedforward networks^{30,33}, one-dimensional convolutional neural networks (1D CNNs) utilize local connections and parameter sharing, resulting in lower computational complexity and reduced risk of overfitting, demonstrated in applications such as electrocardiogram monitoring⁵⁴ and structural damage detection^{55,56}. The Grey Wolf Optimizer (GWO), inspired by the natural behaviors of grey wolf packs⁵⁷, can reliably, quickly, and efficiently explore the search space, making it widely used in optimizing neural network hyperparameters. Three separate 1D CNNs were trained for the extraction of corresponding piezoelectric signal data features and regression identification of the three flight parameters: flapping frequency, wind speed, and pitch angle. We employed GWO to optimize critical parameters for each CNN network (Supplementary Tables 1-3 and Supplementary Movie 6).”

*We added the practical flight test in **Main text** as follows:*

(pages 16-17, lines 455-498) “To substantiate the viability of the bio-hybrid embodied perception approach in practical flight contexts, we meticulously crafted a feathered flapping-wing aircraft (Fig. 8a) capable of untethered flight (Figs. 8b and 8c, Supplementary Movie 11). This aircraft incorporates a gear-driven mechanism, a flight control board, piezoelectric signal acquisition, and high-impedance wireless transmission modules (Supplementary Figs. 33-35). With a total mass of 28.465 g (detailed mass distribution outlined in Supplementary Table 6), the PVDF film represents merely 0.79% of the total mass, whereas the data acquisition and wireless transmission module accounts for 5.15% of the total weight, with the bulk of the mass allocated to the airframe. To refine our mastery over the aircraft's lift and drag dynamics, pivotal for the subsequent development of a perception-motion closed-loop control system, we need discern the relative flow velocities across the flapping wings, the aircraft's pitch angles, and the flapping frequencies. To calibrate the accuracy of embodied sensory identification, we devised a motion capture system (Methods and Supplementary Fig. 36), utilizing an array of eight cameras to concurrently capture the aircraft's trajectory, track marker points, and reconstruct the three-dimensional motion coordinates, thereby computing the instantaneous flight speed and pitch angle for each frame. In indoor windless environments, we posit that the airflow velocity mirrors the aircraft's flying speed in magnitude but is opposite in direction. Direct surveillance of the instantaneous flapping frequency during flight

presents challenges in flapping motion capture. However, we noted a striking concordance between the flapping frequency gauged by an infrared laser displacement sensor and the predominant frequency of the piezoelectric signal when the aircraft engages in wing flapping without actual flight (Supplementary Fig. 37), yielding a relative error of 0.842%. Given that the flight behavior does not alter the correlation between the piezoelectric signal frequency and the actual flapping frequency, we equate the predominant frequency of the piezoelectric signal during flight to the flapping frequency measured by the displacement sensor.

Upon acquisition of the piezoelectric signals via wireless transmission and the corresponding indoor flight data from the motion capture system, these datasets were utilized to constitute a training ensemble (sliding window = 0.01s). The convolutional neural network (CNN) architecture was congruent with the configuration delineated in Fig. 4g. The refined network model was then validated using indoor flight data independent of the training set. The comparison between motion captured values and embodied perceived values is shown in Fig. 8d, with an average absolute error of 0.149 Hz for flapping frequency identification, 0.137 m/s for velocity identification, and 2.360° for pitch angle identification. Our comparative analysis of flapping frequency and velocity demonstrates a high degree of concordance between motion captured and embodied perceived values, underscoring the robust accuracy of our methodologies. In contrast, the detection of pitch angle reveals a slightly elevated error, when compared to non-flight scenarios. We posit that this deviation may stem from the small fluctuations of pitch angle during flight, which complicates the discernment of minute airflow alterations resulting from small directional shifts. Additionally, the concurrent presence of yaw, roll, and flapping motions in the aircraft's operational dynamics contributes to the nuanced challenges in assessing pitch angle through airflow direction and flow-induced vibrations. The intricate aerodynamic interactions and fluid-structure coupling unique to flapping-wing flight, relative to their fixed-wing counterparts, indeed pose a more complex identification challenge. We also conducted outdoor flight tests, applying the CNN model established from indoor flights to identify outdoor flight parameters (Fig. 8e). The practical flight tests of the bio-hybrid flapping-wing robot demonstrate the efficacy of the feather-PVDF embodied perception method, which will aid in the flight state monitoring and reflex control of miniature flapping-wing aircraft, and provide broader insights into the development of embodied perception

in robotics.”

*We rewrote the relevant content in **Discussion** as follows:*

(page 18, lines 502-518) “This research introduces a biohybrid multisensory fusion strategy, uniting natural feathers with flexible piezoelectric films to enable embodied perception in feathered flapping-wing robots. The employed feather-PVDF biohybrid mechanoreceptor closely mirrors avian traits, aiming to maintain natural aerodynamic performance and structural functionality. The study delves into heterointerface connection properties between feathers and PVDF, highlighting excellent peeling strength, durability, and electromechanical performance. The responsive behavior of feather structures to airflow and flapping stimuli translates into distinctive voltage signals on flexible piezoelectric film. Vibration responses of feather branch structures and inter-feather collisions amplify signal differences under diverse stimuli, forming the basis for signal recognition across varied tactile environments and body movements. Regression recognition of voltage signals under different flapping frequencies, headwind speeds, and angles is achieved through CNN and GWO algorithms. The trained network adeptly recognizes temporally varying motion patterns. The biohybrid embodied perception achieves high-precision recognition of the flight state for a flapping-wing robot at an extremely low weight cost. The multisensory fusion perception based on the homology of touch and kinesthesia further enhances the multifunctionality of structural units. Through untethered flight tests, we validated the feasibility of bio-hybrid perception recognition.”

(page 18, lines 519-524) “Biohybrid perception can also be applied to monitor real bird behaviors. Currently, our understanding of bird flight principles is not complete, and direct observation or domestication is time-consuming. Piezoelectric materials have been proven to be biocompatible⁶³. The use of biohybrid perception and wireless transmission technology not only offers new methods for bird monitoring and biological research but also aligns with the principles of biomimetic robotic design.”

*We added the details about indoor and outdoor flight test in **Methods** as follows:*

(page 22, lines 666-682)

“Indoor and outdoor flight test

Indoor flight tests were conducted within a camera cage array, which is a hexadecagon framework

with a side length of 0.98 m, an outer circle diameter of 5.12 m, and a height of 3.52 m. The array was equipped with eight MER-160-227U3M monochrome cameras, each with a resolution of 1440×1080p, featuring Sony IMX273 CMOS sensors with an effective area of 1/1.29 inches. The cameras used Myutron FV0420 lenses with a focal length of 4.16 mm and a maximum aperture of F/2.0. These eight cameras were distributed in a crisscross pattern, with four positioned at a higher level (1.95 m) and four at a lower level (0.40 m), operating at a frame rate of 100 Hz. The cameras were synchronized with the computer and the flight signal acquisition/wireless transmission module. The aircraft was controlled to pass through the array while recording multi-angle video and sensor data of the flight process, detailed processing can be seen in Supplementary Information. Outdoor flight tests were conducted on a spacious lawn under windless conditions. The aircraft was flown for a certain distance, with the flight video recorded using a smartphone camera and flight sensor data collected on a computer. The neural network structure is consistent with Fig. 4g, and the training set is derived from the data collected from indoor flight, and the identification performance of the three networks is tested using the complete indoor flight data independent of the training data set.”

Fig. 8: Flight testing of the flapping-wing robot with feathered wings.

a Structural components of the flapping-wing robot with feathered wings. **b** Indoor flight, the data independent of the training set is used as the indoor flight test data to obtain the corresponding network model. **c** Outdoor flight, using the verified network model in Fig. **b** for identification and display. **d** Comparison of the test and perception values in Fig. **b**, the identification error distributions of flapping frequency (blue), velocity (orange) and pitch angle (green) are shown on the right side, the box plots show the maximum, minimum, 25th percentile, 75th percentile, and median error values for each category, the black horizontal line represents the perception average error value. **e** The perception results of outdoor flight data in Fig. **c** are displayed.

Other specific comments:

1. The detailed sensor structure and sensing principle of the PVDF is not clear. There seem to be two parallel silver electrodes from Fig. 2. Is this based on parallel capacitive sensing? If this is the case, the novelty of this key aspect is insufficient as such kinds of sensors have been very commonly reported.

Response: Thank you for your comments. Current research on flight parameter sensing and recognition predominantly focuses on fixed-wing and rotary-wing domains, achieving significant advancements in miniaturization of vision-based flight control and inertial measurement units (IMUs). Inspired by the airflow sensing mechanisms of birds, bats, and other biological systems, the use of pressure and flow sensors to estimate multiple flight parameters has been proposed as a viable solution for intelligent flight control [1]. For biomimetic flapping-wing aircraft, the stringent weight constraints and complex dynamics and control systems impose higher demands on lightweight, integrated, and highly sensitive flexible sensors. Piezoelectric sensing, known for its reproducibility and practical application value, is extensively used in tactile sensing, biomedical sensing, and wearable flexible electronics. PVDF (polyvinylidene fluoride) piezoelectric polymers, with their low density, high flexibility, and ability to adapt to various shapes, are particularly suited for flapping-wing aircraft with strict weight limitations, as they can operate passively without external power sources. The PVDF piezoelectric film we use consists of a five-layer structure: two polypropylene layers serving as protective layers, two silver-coated layers functioning as electrodes, and one core PVDF layer providing the piezoelectric conversion function. The silver electrode layers form a conductive network through uniformly distributed silver particles, ensuring balanced charge loading and minimizing loss during charge accumulation and transmission. The core PVDF layer, a polycrystalline functional polymer material, exhibits excellent piezoelectric properties and is only 28 micrometers thick. The piezoelectric sensing principle employed is the positive piezoelectric effect, where applying physical pressure to the piezoelectric material shortens the dipole moments within the material. To counteract this change, the material generates equal and opposite charges on its surfaces, causing electron movement and forming a current. Essentially, the positive piezoelectric effect is the conversion of mechanical energy into electrical energy. Therefore, using PVDF as a sensor allows it to function without external power, conserving energy for the flapping-wing aircraft system. We have added illustrative diagrams and detailed explanations of the piezoelectric film's working principle in the supplementary materials to clearly elucidate its operation mechanism.

It is worth further elucidating that the bio-hybrid sensing method proposed in this study comprises two components: the feather-based vibration structure that transmits deformation

signals and the piezoelectric signal generator for electromechanical conversion. The unique structure and material properties of feathers endow them with excellent aerodynamic performance, one of the secrets behind birds' exceptional flight capabilities. Another secret lies in the wings' sensing capabilities. Studies have shown that most feathers exhibit different vibration modes and frequencies under varying airflow speeds and directions, and adjacent vibrating feathers can act as coupled oscillators, producing collisions and resulting in nonlinear interactions [2]. By utilizing the natural structure of feathers as the foundational structure for vibration and airflow sensing, and leveraging the differential deformation of flexible PVDF piezoelectric films that conform to feather movements, we can source variations in piezoelectric signals. This approach retains the biomechanical value in the sensing structure of flapping-wing aircraft. Unlike improvements to the flexible sensor structure itself, our bio-hybrid sensing method enhances mechanical perception through natural biological materials and structures, combined with the lightweight, self-powered PVDF to mimic the mechanoreceptor functions of bird wings. This study explores the adhesion mechanics of heterogeneous interfaces and the electromechanical conversion performance from both mechanical and electromechanical perspectives. We aim to achieve high-precision, multi-parameter recognition of the biomimetic flapping-wing aircraft through bio-hybrid sensing. The supplementary real-flight sensing demonstration experiments further enhance the applicability and persuasiveness of this sensing technology, promoting the development of lighter and more integrated flapping-wing aircraft. This point has also been supplemented in the main text to provide readers with a clearer understanding of the bio-hybrid embodied sensing concept proposed in this study.

[1] Gong, Z. et al. Flexible calorimetric flow sensor with unprecedented sensitivity and directional resolution for multiple flight parameter detection. *Nat. Commun.* **15**, 3091 (2024).

[2] Clark, C. J., Elias, D. O. & Prum, R. O. Aeroelastic Flutter Produces Hummingbird Feather Songs. *Science*. **333**, 1430-1433 (2011).

*We added the development limitations of flapping wing sensing in **Introduction** as follows:*

(page 2, lines 63-65) “Presently, flexible sensing in unmanned aerial vehicles mainly focuses on fixed-wing configurations²⁹⁻³³. For flapping-wing aircraft, which have more complex power principles and stricter weight limitations, flexible embodied perception in actual flight remains

challenging³⁴⁻³⁶.”

*We added the discussion of feathers in **Introduction** as follows:*

(page 2, lines 68-77) “Research indicates that most feathers exhibit distinct vibration modes and frequencies under varying flow velocities and directions. The adjacent vibratory feathers might operate as coupled oscillators, culminating in collisions that engender nonlinear interactions⁴². Feathers exhibit remarkable responsiveness to external airflow while boasting lightweight, tear-resistant characteristics^{43,44}. Moreover, they possess a restitution property, enabling their surface morphology to be restored following disorder, merely through the application of gentle stroking or tidying. Drawing upon natural feather biomaterials and structures to enhance the differentiation of mechanical perception, integrating flexible lightweight self-powered PVDF to emulate the functions of avian wing mechanoreceptors, this study proposes a biohybrid embodied perception approach with deep learning (Fig. 1a and Supplementary Movie 1).”

*We rewrote the relevant content in **Discussion** is as follows:*

(page 18, lines 502-518) “This research introduces a biohybrid multisensory fusion strategy, uniting natural feathers with flexible piezoelectric films to enable embodied perception in feathered flapping-wing robots. The employed feather-PVDF biohybrid mechanoreceptor closely mirrors avian traits, aiming to maintain natural aerodynamic performance and structural functionality. The study delves into heterointerface connection properties between feathers and PVDF, highlighting excellent peeling strength, durability, and electromechanical performance. The responsive behavior of feather structures to airflow and flapping stimuli translates into distinctive voltage signals on flexible piezoelectric film. Vibration responses of feather branch structures and inter-feather collisions amplify signal differences under diverse stimuli, forming the basis for signal recognition across varied tactile environments and body movements. Regression recognition of voltage signals under different flapping frequencies, headwind speeds, and angles is achieved through CNN and GWO algorithms. The trained network adeptly recognizes temporally varying motion patterns. The biohybrid embodied perception achieves high-precision recognition of the flight state for a flapping-wing robot at an extremely low weight cost. The multisensory fusion perception based on the homology of touch and kinesthesia further enhances the multifunctionality of structural units.

Through untethered flight tests, we validated the feasibility of bio-hybrid perception recognition.”

We added the working principle of piezoelectric materials in *supplementary materials* as follows:

(page 5, lines 113-127) **“1. The working principle of piezoelectric materials**

When a piezoelectric material undergoes deformation due to external forces, polarization occurs internally. This excites movement of positive and negative charge centers within the material, resulting in opposite polarity charges of equal magnitude on its upper and lower surfaces, and generating an electric field within. Upon removal of the external force, the piezoelectric material gradually returns to its original undeformed state, causing the electric field strength to diminish until the charged state disappears. Reversal of external force causes a corresponding reversal in the movement of positive and negative charges. Piezoelectric sensors utilize the direct piezoelectric effect principle, where electrodes connected to the upper and lower surfaces of the material form a circuit. When the sensor undergoes deformation due to external forces, voltage changes can be detected to reflect the applied load (Supplementary Fig. 1).”

Supplementary Fig. 1: Schematic illustration of the working principle of piezoelectric thin films.

2. Fig. 4 i, There seem to be considerable errors, especially for wind speed and pitch angle estimation. For some results, overlapping between levels is evident. There should be criteria regarding the usability of such results.

Response: Thank you for your comments. In response to the results presented in Fig. 4i, we retrained the model and utilized a box plot to represent the absolute error results of the test set, replacing the original Fig. 4i for a more intuitive illustration of error levels. The box plot displays the maximum, minimum, 25th percentile, 75th percentile, and median values of the errors for each category. The triangle points denote the mean error for each category, while the dashed line indicates the mean error across all test samples.

We focused on the mean absolute error (MAE) metric for the sample points, as this parameter directly reflects the recognition accuracy. The average recognition error for flapping frequency (0-6Hz) is 0.043Hz, for wind speed (0-4.35m/s) is 0.064m/s, and for pitch angle (5-45°) is 0.910°. Although some categories exhibit individual outliers in absolute error, the overall mean error remains at a relatively low level. Additionally, we have included a comparison of our work with the recognition errors of other fixed-wing aircraft in the supplementary materials. Despite the more complex dynamics of flapping wings compared to fixed wings, the comparative data clearly indicate that our work achieves recognition accuracy comparable to that of fixed-wing aircraft. It is noteworthy that we employed only one single bio-hybrid sensor, which significantly simplifies the sensing system.

*We added the description of the platform training results in **Main text** as follows:*

(page 11, lines 315-325) “We calculated the absolute errors and data distributions in the test set (Fig. 4h), yielding an average absolute error of 0.043 Hz for flapping frequency perception, 0.064 m/s for wind speed perception, and a root mean square error of 0.910° for pitch angle perception. Despite occasional outliers in absolute errors for specific categories, the overall average error remains at a low level. The relative error distributions and iteration process for the identification of the three parameters are shown in Supplementary Figs. 25 and 26, indicating excellent regression fitting accuracy of the trained networks. We contrast the perception errors of our flapping-wing flight against fixed-wing flight^{33,58-61} (Supplementary Table 4), evidencing either equivalent or elevated recognition fidelity, especially in the context of velocity identification. It is noteworthy to point out that our approach utilizes a single bio-hybrid sensor, effectuating a notable diminution in both complexity and weight for the sensor assembly.”

Fig. 4: Feather-PVDF biohybrid mechanoreceptor for embodied energy and voltage signal acquisition, and convolutional neural network-grey wolf optimizer algorithm for embodied perception recognition.

h Box plots of the absolute errors in the identification of flapping frequency (blue), wind speed (yellow), and pitch angle (red). The box plots show the maximum, minimum, 25th percentile, 75th percentile, and median error values for each category. Triangular points represent the mean error for each category, while the dashed line indicates the mean error across all test samples.

We added the comparative results in *supplementary materials* as follows:

(page 43, lines 822-824) **Supplementary Table 4. Comparative analysis of parameter identification error in our work versus other fixed-wing studies**

Sensor type	Sensor num	Velocity error	Flapping frequency error	Angle error	Aircraft type	Ref
Pressure sensors	24	0.459m/s (12-18 m/s)	-	0.340° (AOA:-3~11°)	Fixed wing	9
	5	0.620m/s (12-20 m/s)	-	0.510° (AOA:-9~11°)	Fixed wing	10
	4	0.758m/s (7-15 m/s)	-	1.792° (AOA:-4~16°)	Fixed wing	11
Flow sensors	3	0.270m/s (5-28 m/s)	-	0.870° (AOA:0~20°)	Fixed wing	12
	2	0.151m/s (2-4 m/s)	-	0.580° (AOA:-20~20°)	Fixed wing	13
Biohybrid sensor	1	0.064m/s (1-4.35m/s)	0.043Hz	0.910° (Pitch:5~45°)	Flapping wing	Our

AOA: Angle of Attack

9. Li, N. et al. A compact embedded flight parameter detection system for small soaring UAVs. *IEEE-ASME Trans. Mechatron.* **29**, 52-63 (2024).
10. Samy, I., Postlethwaite, I., Gu, D. & Green, J. Neural-network-based flush air data sensing

system demonstrated on a mini air vehicle. *J. Aircr.* **47**, 18-31 (2010).

11. Wood, K. T., Araujo-Estrada, S., Richardson, T. & Windsor, S. Distributed pressure sensing - based flight control for small fixed-wing unmanned aerial systems. *J. Aircr.* **56**, 1951-1960 (2019).
12. Zhu, R., Que, R. & Liu, P. Flexible micro flow sensor for micro aerial vehicles. *Front. Mech. Eng.* **12**, 539-545 (2017).
13. Gong, Z. et al. Flexible calorimetric flow sensor with unprecedented sensitivity and directional resolution for multiple flight parameter detection. *Nat. Commun.* **15**, 3091 (2024).

3. Line 81, what is PVDF?

Response: Thank you for your question. PVDF is the abbreviation of polyvinylidene fluoride. We have supplemented the full term at the first instance where the abbreviation was used.

We added the relevant content in **Main text** as follows:

(page 2, lines 58-60) “Lightweight, sensitive, and adherent flexible Polyvinylidene Fluoride (PVDF) piezoelectric films show promise in effectively reflecting pressure and vibration behaviors²³⁻²⁶”

4. The detailed architecture of the network is not clear.

Response: Thank you for your comments. We employed a one-dimensional convolutional neural network (1D CNN) to analyze the voltage signals generated by the feather-PVDF intelligent material under varying flapping frequencies, wind speeds, and angles of attack. The data were segmented using a 1-second window as the optimal input size (as detailed in the supplementary materials). All samples were randomly divided into training (80%), testing (10%), and validation (10%) sets for network training and evaluation. The initial data were normalized and converted into the input format $1000 \times 1 \times 1 \times N$ (where N is the number of samples) for training. The network comprises three convolutional layers, three pooling layers, and three fully connected layers, with the specific stacking order as follows: data input → convolutional layer → loss function (ReLU) → pooling layer → convolutional layer → loss function (ReLU) → pooling layer → convolutional layer → loss function (ReLU) → pooling layer → fully connected layer (3 layers) → regression layer. Detailed parameters are provided in the supplementary materials.

We added the description of the network structure in **supplementary materials** as follows:

(page 39, lines 763-770) “The initial data were normalized and transformed into the input format of $1000 \times 1 \times 1 \times N$ (N is the number of samples) for training. The network structure is illustrated in Figure 4g, comprising three convolutional layers, three pooling layers, and three fully connected layers, with the specific stacking sequence as follows: data input \rightarrow convolutional layer \rightarrow loss function (ReLU) \rightarrow pooling layer \rightarrow convolutional layer \rightarrow loss function (ReLU) \rightarrow pooling layer \rightarrow convolutional layer \rightarrow loss function (ReLU) \rightarrow pooling layer \rightarrow fully connected layers (3 layers) \rightarrow regression layer.”

(page 40, lines 786-788) **Supplementary Table 1. The hyperparameters of the neural network for flapping frequency recognition**

No	Layer Type	Kernel size	Kernels number	Pool Size	Stride	FC Size	Learning rate
1	Conv_1	206	143				0.0017
2	Maxpool_1			9	25		
3	Conv_2	231	38				
4	Maxpool_2			10	2		
5	Conv_1	39	24				
6	Maxpool_1			1	1		
7	FC					44	
8	FC					17	
9	FC					1	
10	Regression						

(pages 40-41, lines 789-791) **Supplementary Table 2. The hyperparameters of the neural network for oncoming flow velocity recognition**

No	Layer Type	Kernel size	Kernels number	Pool Size	Stride	FC Size	Learning rate
1	Conv_1	184	197				6.11e-04

2	Maxpool_1			82	16		
3	Conv_2	148	159				
4	Maxpool_2			13	6		
5	Conv_1	1	206				
6	Maxpool_1			1	32		
7	FC						12
8	FC						16
9	FC						1
10	Regression						

(page 41, lines 792-794) **Supplementary Table 3. The hyperparameters of the neural network for pitch angle recognition**

No	Layer Type	Kernel size	Kernels number	Pool Size	Stride	FC Size	Learning rate
1	Conv_1	145	188				0.0016
2	Maxpool_1			97	13		
3	Conv_2	66	224				
4	Maxpool_2			21	8		
5	Conv_1	48	143				
6	Maxpool_1			2	6		
7	FC					43	
8	FC					45	
9	FC					1	
10	Regression						

5. Fig.2 j, it is not visible regarding the detailed waveforms of the cycling test. Is that a linear response or a nonlinear one?

Response: Thank you for your comments. We have added detailed waveforms of the three stages of the vibration experiment of a single feather with tip mass in Fig. 2j. Additionally, we

conducted a frequency spectrum analysis of the voltage signal in 10,000 cycles. The frequency spectrum demonstrates a primary frequency at 30 Hz, which is also the excitation frequency, without any extraneous frequency components, indicating a linear response of a single feather with tip mass under the above forced vibration conditions.

We added signal analysis in **Main text** as follows:

(page 6, lines 150-155) “Normalized voltage curves (Fig. 2j) show that during 10,000 vibration tests, the structural output performance exhibits no significant change. Analysis of local signals and spectra at different stages (Supplementary Fig. 5) display evident single-period characteristics, demonstrating the durable electromechanical conversion performance of the feather-PVDF biohybrid mechanoreceptor even after prolonged fatigue testing.”

We added the details of the signal in **supplementary materials** as follows:

(page 10, lines 211-216) “**3.Signals analysis during dynamic fatigue testing**”

Supplementary Fig. 5: Localized waveform and spectral analysis of voltage during dynamic fatigue testing. **a** Localized voltage signals in the initial (11-15), intermediate (5261-5265), and terminal (9995-9999) cycling tests. **b** Spectral analysis of normalized voltage signals in 10,000 cycles of loading.

6. Most of the figures are of low quality/resolution, hindering information extraction.

Response: Thank you for your comment. We have taken particular care to address this issue

during this submission and have enhanced the clarity of the images accordingly.

Reviewer #1 (Remarks on code availability):

I don't have access to any codes.

Response: Thank you for your comment. We will upload the codes upon acceptance of the manuscript.

Reviewer #2

Interesting work, but there is a mismatch between the title and the content of the article. Specifically the title refers to "flapping-wing robotics" while the body of the text only shows flapping devices used to test the hybrid systems without any demonstration of flight. As written, the article is not suitable for publication in Nature Communications, as it demonstrates development of a sensor and some data processing methods.

For the work to be suitable, it would need to grow in one of two possible directions:

-demonstrate the device in an actual flapping wing robot capable of lift, even if tethered to a power source. The data acquired during this type of demonstration would be a significant proof of the sensing method and its value to the broader community.

-drop the focus on robotics, and broaden the bio-inspired perspective. One suggestion is to look at the mechanosensory capability of fishes using their lateral line system, one good reference to start with is Webb, J.F., Maruska, K.P., Butler, J.M. and Schwalbe, M.A., 2021. The mechanosensory lateral line system of cichlid fishes: from anatomy to behavior. The behavior, ecology and evolution of cichlid fishes, pp.401-442. The parallels between lateral line and feather sensing in moving fluids would be a great source of inspiration across disciplines, from biology to engineering.

Response: We sincerely appreciate your positive feedback and valuable suggestions on our research work. In response to your recommendations for improvement, we have developed a customized feathered flapping-wing aircraft to validate the bio-hybrid embodied sensing approach in real-flight scenarios. The aircraft, weighing 28.465 grams, includes a gear-driven module, flight control board, and piezoelectric signal acquisition with high-impedance wireless transmission. A motion capture system with eight cameras was used to measure flight parameters for validation. The aircraft demonstrated untethered flight, with a motion capture system tracking its flight to calculate speed and pitch angle. Piezoelectric signals and flight data were used to train a CNN model, which was then validated with independent indoor flight data, showing an average absolute error of 0.149 Hz for flapping frequency, 0.137 m/s for velocity, and 2.360 ° for pitch angle. Outdoor flight tests further validated the CNN model's effectiveness. This bio-hybrid sensing method enhances flight state monitoring and reflex control for miniature flapping-wing aircraft, offering broader insights into embodied sensing in robotics.

*We added the practical flight test in **Main text** as follows:*

(pages 16-17, lines 455-498) “To substantiate the viability of the bio-hybrid embodied perception approach in practical flight contexts, we meticulously crafted a feathered flapping-wing aircraft (Fig. 8a) capable of untethered flight (Figs. 8b and 8c, Supplementary Movie 11). This aircraft incorporates a gear-driven mechanism, a flight control board, piezoelectric signal acquisition, and high-impedance wireless transmission modules (Supplementary Figs. 33-35). With a total mass of 28.465 g (detailed mass distribution outlined in Supplementary Table 6), the PVDF film represents merely 0.79% of the total mass, whereas the data acquisition and wireless transmission module accounts for 5.15% of the total weight, with the bulk of the mass allocated to the airframe. To refine our mastery over the aircraft's lift and drag dynamics, pivotal for the subsequent development of a perception-motion closed-loop control system, we need discern the relative flow velocities across the flapping wings, the aircraft's pitch angles, and the flapping frequencies. To calibrate the accuracy of embodied sensory identification, we devised a motion capture system (Methods and Supplementary Fig. 36), utilizing an array of eight cameras to concurrently capture the aircraft's trajectory, track marker points, and reconstruct the three-dimensional motion coordinates, thereby computing the instantaneous flight speed and pitch angle for each frame. In indoor windless environments, we posit that the airflow velocity mirrors the aircraft's flying speed in magnitude but is opposite in direction. Direct surveillance of the instantaneous flapping frequency during flight presents challenges in flapping motion capture. However, we noted a striking concordance between the flapping frequency gauged by an infrared laser displacement sensor and the predominant frequency of the piezoelectric signal when the aircraft engages in wing flapping without actual flight (Supplementary Fig. 37), yielding a relative error of 0.842%. Given that the flight behavior does not alter the correlation between the piezoelectric signal frequency and the actual flapping frequency, we equate the predominant frequency of the piezoelectric signal during flight to the flapping frequency measured by the displacement sensor.

Upon acquisition of the piezoelectric signals via wireless transmission and the corresponding indoor flight data from the motion capture system, these datasets were utilized to constitute a training ensemble (sliding window = 0.01s). The convolutional neural network (CNN) architecture was congruent with the configuration delineated in Fig. 4g. The refined network model was then

validated using indoor flight data independent of the training set. The comparison between motion captured values and embodied perceived values is shown in Fig. 8d, with an average absolute error of 0.149 Hz for flapping frequency identification, 0.137 m/s for velocity identification, and 2.360° for pitch angle identification. Our comparative analysis of flapping frequency and velocity demonstrates a high degree of concordance between motion captured and embodied perceived values, underscoring the robust accuracy of our methodologies. In contrast, the detection of pitch angle reveals a slightly elevated error, when compared to non-flight scenarios. We posit that this deviation may stem from the small fluctuations of pitch angle during flight, which complicates the discernment of minute airflow alterations resulting from small directional shifts. Additionally, the concurrent presence of yaw, roll, and flapping motions in the aircraft's operational dynamics contributes to the nuanced challenges in assessing pitch angle through airflow direction and flow-induced vibrations. The intricate aerodynamic interactions and fluid-structure coupling unique to flapping-wing flight, relative to their fixed-wing counterparts, indeed pose a more complex identification challenge. We also conducted outdoor flight tests, applying the CNN model established from indoor flights to identify outdoor flight parameters (Fig. 8e). The practical flight tests of the bio-hybrid flapping-wing robot demonstrate the efficacy of the feather-PVDF embodied perception method, which will aid in the flight state monitoring and reflex control of miniature flapping-wing aircraft, and provide broader insights into the development of embodied perception in robotics.”

Fig. 8: Flight testing of the flapping-wing robot with feathered wings.

a Structural components of the flapping-wing robot with feathered wings. **b** Indoor flight, the data independent of the training set is used as the indoor flight test data to obtain the corresponding network model. **c** Outdoor flight, using the verified network model in Fig. **b** for identification and display. **d** Comparison of the test and perception values in Fig. **b**, the identification error distributions of flapping frequency (blue), velocity (orange) and pitch angle (green) are shown on the right side, the box plots show the maximum, minimum, 25th percentile, 75th percentile, and median error values for each category, the black horizontal line represents the perception average error value. **e** The perception results of outdoor flight data in Fig. **c** are displayed.

We added the details about indoor and outdoor flight test in **Methods** as follows:

(page 22, lines 666-682)

“Indoor and outdoor flight test

Indoor flight tests were conducted within a camera cage array, which is a hexadecagon framework with a side length of 0.98 m, an outer circle diameter of 5.12 m, and a height of 3.52 m. The array was equipped with eight MER-160-227U3M monochrome cameras, each with a resolution of 1440×1080p, featuring Sony IMX273 CMOS sensors with an effective area of 1/1.29 inches. The cameras used Myutron FV0420 lenses with a focal length of 4.16 mm and a maximum aperture of

F/2.0. These eight cameras were distributed in a crisscross pattern, with four positioned at a higher level (1.95 m) and four at a lower level (0.40 m), operating at a frame rate of 100 Hz. The cameras were synchronized with the computer and the flight signal acquisition/wireless transmission module. The aircraft was controlled to pass through the array while recording multi-angle video and sensor data of the flight process, detailed processing can be seen in Supplementary Information. Outdoor flight tests were conducted on a spacious lawn under windless conditions. The aircraft was flown for a certain distance, with the flight video recorded using a smartphone camera and flight sensor data collected on a computer. The neural network structure is consistent with Fig. 4g, and the training set is derived from the data collected from indoor flight, and the identification performance of the three networks is tested using the complete indoor flight data independent of the training data set.”

*We added the details of the robot construction and flight experiments in **supplementary materials** as follows:*

(pages 51-54, lines 948-1023)“**12. Practical flight experiments**

The motion capture system used in our indoor experiments has a hexadecagon framework with a side length of 0.98 m, a circumcircle diameter of 5.12 m, and a height of 3.52 m, with a green screen used for external coverage. Eight MER-160-227U3M black-and-white cameras were used for capturing, each with a resolution of 1440×1080p, equipped with Sony IMX273 CMOS sensors, and having an effective area of 1/1.29 inches. The cameras were fitted with Myutron FV0420 lenses, featuring a focal length of 4.16 mm and a maximum aperture of F/2.0. The eight cameras were evenly distributed in a crossed pattern, with four positioned at a higher level (1.95 m) and four at a lower level (0.40 m), operating at a frame rate of 100 Hz.

During actual flight, due to the frame rate limitations, it is challenging to precisely identify instantaneous flapping frequencies. Therefore, we collected the voltage signals generated during flight and performed FFT (Fast Fourier Transform) on these signals to determine the primary frequency, which was then used as the flapping frequency at that moment. The flight speed was provided by the motion capture system. Using the intrinsic and extrinsic parameters obtained through calibration of the cameras, we placed white markers on the head and tail of the flying device.

The position of these markers in each frame of the flight video was recorded using the MATLAB APP DLTdv8a, allowing for the reconstruction of the three-dimensional space. By obtaining the displacement data of the head and tail markers in the X, Y, and Z directions and calculating the pitch angle and displacement increments per unit time, we determined the speed, with the unit time being the reciprocal of the frame rate. The time series pitch Angle and velocity data are processed by moving mean and used as the final test values. During indoor flights, the aircraft performs maneuvers such as turning and ascending, occasionally moving out of the camera array's capture range. Consequently, the tested values for flight speed and pitch angle may exhibit interruptions. We selected flight samples that were fully captured by the camera array as our training set and used data independent of the training set as the test set for perception validation.”

“We have developed a customized ornithopter featuring a wing-flapping mechanism. The wings are constructed with a PET film as the primary support layer, with PVDF-feather hybrid sensors mounted on the outermost edge of each wing. Additional feathers are layered sequentially over the PET base. Lightweight clasps are positioned along the wing's central axis to ensure secure attachment and stability during flapping. Wing motion is driven by a gear assembly and a miniature motor.

A signal acquisition/wireless transmission module, measuring $2.2 \text{ cm} \times 1.6 \text{ cm} \times 1.6 \text{ mm}$ and weighing 1.465 g, is mounted at the rear of the ornithopter and connected to the PVDF-feather hybrid sensors. Both the flight control board and the signal acquisition/wireless transmission module are powered by a single 1s lithium battery, with the entire robot weighing 28.465 g. This setup enables untethered flight, as well as signal acquisition and wireless transmission. The overall operational principle is illustrated in the Supplementary Fig.33. The first part is the flight drive module, which includes the flight control board, micro motors, and gear set to control the wing flapping frequency. The specific component connection diagram is shown in Supplementary Fig.34. The electronic components on the flight control board include an MCU (PAN742), IMU (BMI270), Low-Dropout Regulator, and a wireless transceiver module (CC2530), all powered by a 100mAh 1S lithium battery. The second part is the biohybrid sensing module, which comprises the biohybrid sensor, signal acquisition/wireless transmission circuit board, and CNN algorithm to enable the

recognition of flight parameters. This module is also powered by the same 100mAh 1S lithium battery. The specific component connection diagram is shown in Supplementary Fig.35. The electronic components on the signal acquisition/wireless transmission board include an MCU (STM32), Low-Dropout Regulator, a wireless transceiver module (CH9140), operational amplifiers, and switched capacitor inverter. The sensor is connected to the signal acquisition/wireless transmission board via wires. Bluetooth communication is achieved using BleComManager software, with the host computer sending commands for data acquisition and wireless transmission. The sensor data is then processed by the CNN algorithm module for recognition.”

Supplementary Fig. 33: Operational principles of the flapping-wing robot.

Supplementary Fig. 34: Schematic diagram of the component connections in the drive module.

Supplementary Fig. 35: Integrated signal acquisition and wireless transmission module. a Display of modules. b Schematic diagram of the component connections in the perception module.

Supplementary Table 6. Mass decomposition chart for feathered flapping-wing robot

Compositional parts	Mass(g)	Percentage
PVDF	0.225	0.79%
Signal acquisition and wireless transmission module	1.465	5.15%
Conducting wire	1.834	6.44%
Feathered wings	4.449	15.63%
Fuselage (skeleton + flight control board + power supply)	20.492	71.99%
Total	28.465	100.00%

Supplementary Fig. 36: The motion capture system.

Supplementary Fig. 37: Comparison of flapping frequency monitored by the laser displacement sensor with the signal's dominant frequency, comprising 298 samples, showing the average relative error of 0.842%.

General suggestion to improve the figures:

-the text in Fig. 4, 5, 6 and 7 is difficult to read, the resolution of the figures is poor.

Response: Thank you for your comments. In the current submission, we have paid special attention to this issue and have enhanced the resolution of the figures accordingly.

*We showed the relevant pictures in **Main text** as follows:*

Fig. 4: Feather-PVDF biohybrid mechanoreceptor for embodied energy and voltage signal acquisition, and convolutional neural network-grey wolf optimizer algorithm for embodied perception recognition.

Fig. 5: Experiment of frequency-varying motion signal recognition in continuous time.

Fig. 6: Recognition of flapping frequency motion and wing morphology in the feathered flapping wing robot.

Fig. 7: Real-time recognition process, results, and visualization interface of feathered flapping wing robot.

-the difference between c and d in Figure 2 is not immediately apparent, could be clarified.

Response: Thank you for your comment. Fig. 2c and 2d respectively present the cross-section morphology characteristics of the feather-PVDF piezoelectric sensor in the transverse (perpendicular to the feather shaft) and longitudinal (parallel to the feather shaft) directions. Along each direction, we provide the adhesive observation at randomly selected three locations. To more intuitively distinguish between the two subfigures, we have included the cutting direction indications in the top right corner of Fig. 2c and 2d.

We revised the relevant pictures in Main text as follows:

Fig. 2: Characterization and performance testing of feather-PVDF biohybrid mechanoreceptor. **c** Microscopic characterization for the adhesive cross-section of feather-PVDF along the blue dashed line perpendicular to the feather rachis in the upper right corner of the figure. **d** Microscopic characterization for the adhesive cross-section of feather-PVDF along the orange dashed line parallel to the feather rachis in the upper right corner of the figure.

-the purpose of figures h and i in the same figure is unclear, because it doesn't compare systems with and without PVDF

Response: Thank you for your comment. We provided the microscopic characterization of the feather to demonstrate that the fibrous barbs on the feather surface can achieve good adhesion with the PVDF film. Based on your suggestion, we used a low-vacuum ultra-high-resolution field emission scanning electron microscope to capture the adhesion condition of the feather-PVDF interface. Fig. 2e presents the microscopic characterization of the cross section perpendicular to the feather shaft, clearly illustrating tight adhesion between the interface and interlocking by barbicels of barbules.

We added the microscopic characterization in **Main text** as follows:

(page 5, lines 142-147)“PET and carbon fiber exhibit smoother continuous surfaces, while feathers are covered in fine discrete barbs (Fig. 2h). Similar fibrous features are commonly found in natural biological structures, such as the foot pads of geckos⁴⁵. The fibrous features contribute to increased adhesion in contact with the film. At the micron scale, we observed the cross section of heterogeneous interface between the feather and PVDF (Fig. 2i and Supplementary Fig. 4), which exhibited tight adhesion between the interface and interlocking by barbicels of barbules.”

Fig. 2: Characterization and performance testing of feather-PVDF biohybrid mechanoreceptor. h Microstructure of the feather surface. **i** Microstructure of the heterogeneous attachment cross-section between the feather and PVDF.

We added the microscopic characterization in supplementary materials as follows:

(page 9, lines 192-196)

Supplementary Fig.4: Microscopic characterization of feather-PVDF adhesion. a-b Microscopic characterization of the cross-sectional adhesion. **c** Microscopic characterization of the adhesion surface. **d** Local magnification of the adhesion surface microscopic features.

-adding sensing capability to feathers is one thing, however the major advantage in natural systems is the ability to rapidly change between wing shapes, as shown in Figure 6 f. Does the specific PVDF method lead the authors to believe one specific morphing wing technology is more suitable for feathers embedded with this type of sensing? What downstream design

impact does adding the PVDF and learning algorithm have on robot design?

Response: Thank you for your comments. Our sensing method is compatible with the continuous morphing wing technology composed of discrete units [1]. The interlocking characteristics, self-healing properties, and lightweight nature of natural feathers make them ideal for use as morphing units. By using servo-driven wrist and finger joints to control wings composed of discrete feather units, adjacent units are constrained by elastic cords to achieve continuous deformation [2]. The frictional interlocking between feathers allows for smoother and more stable deformations. These elastic cord connection, combined with the frictional interlocking of the feathers, prevent the feathers from dispersing and ensure precise unit movement. This morphing wing technology has been demonstrated to facilitate agile and flexible wing deformations, enhancing maneuverability. Our feather-PVDF sensing method using a single flexible PVDF film on the surface of a single uppermost feather, ensuring it does not interfere with the frictional interlocking of the underlying feathers and elastic cord constrain between the feathers. Thus, our proposed bio-hybrid sensing method can be combined with the morphing wing method using elastic cord constraints and frictional interlocking, as described in reference [2], to achieve wing shape sensing in morphing wings.

Biomimetic robotic design draws inspiration from natural organisms and biological materials, celebrated for their intricately interconnected systems capable of diverse functionalities. The integration of intelligent and biomimetic materials offers insights into the embodied intelligence design of flapping-wing robots. Our feather-PVDF mechanoreceptor is particularly well-suited for discrete morphing wings with elastic tendons and frictional hooking, fostering potential synergistic advancements in feathered flapping-wing robotics. With the continuous advancement of artificial intelligence models, multimodal embodied intelligence emerges as an effective design paradigm in the realm of robotics. This approach underscores the "perception-action loop" and introduces a "multi-level evolution" design framework for materials and machines. By integrating interactions between the body and the environment to obtain sensory feedback, thereby serving the control system, an effective neural control strategy is established. However, the practical integration of these biohybrid perception functions into reflex-based control will necessitate extensive flight testing and the development of lightweight

data collection, transmission systems, and responsive multifunctional actuators. Furthermore, achieving autonomous interaction between FWRs and the environment, fostering adaptive intelligent behavior, will require more efficient and stable intelligent algorithms. This trend will further propel the integration of multifunctionality and intelligence in future autonomous robotic systems, encompassing aspects of sensing, decision-making, response, and energy.

[1] Chang, E., Matloff, L. Y., Stowers, A. K. & Lentink, D. Soft biohybrid morphing wings with feathers underactuated by wrist and finger motion. *Sci. Robot.* **5**, 1246 (2020).

[2] Matloff, L. Y. et al. How flight feathers stick together to form a continuous morphing wing. *Science.* **367**, 293 (2020).

*We added the discussion of design paradigm for robots in **Discussion** as follows:*

(page 18, lines 524-541) “Biomimetic robotic design draws inspiration from natural organisms and biological materials, celebrated for their intricately interconnected systems capable of diverse functionalities⁶⁴. The integration of intelligent and biomimetic materials offers insights into the embodied intelligence design of flapping-wing robots. Our feather-PVDF mechanoreceptor is particularly well-suited for discrete morphing wings with elastic tendons and frictional hooking, fostering potential synergistic advancements in feathered flapping-wing robotics^{5,11}. With the continuous advancement of artificial intelligence models, multimodal embodied intelligence emerges as an effective design paradigm in the realm of robotics⁶⁵. This approach underscores the "perception-action loop" and introduces a "multi-level evolution" design framework for materials and machines⁶⁶. By integrating interactions between the body and the environment to obtain sensory feedback, thereby serving the control system, an effective neural control strategy is established. However, the practical integration of these biohybrid perception functions into reflex-based control will necessitate extensive flight testing^{11,12,67} and the development of lightweight data collection, transmission systems, and responsive multifunctional actuators. Furthermore, achieving autonomous interaction between FWRs and the environment, fostering adaptive intelligent behavior⁶⁸, will require more efficient and stable intelligent algorithms. This trend will further propel the integration of multifunctionality and intelligence in future autonomous robotic systems, encompassing aspects of sensing, decision-making, response, and energy.”

Reviewer #3

The manuscript reports on experiments with a flapping wing robot with piezoelectric PVDF films integrated into its feathered wings. The authors report on the manufacturing processes of the wings and especially the film bonding procedure. Experiments at various flapping frequencies, pitch angles, wind velocities and feather arrangements were conducted using the tethered robot. Employing Convolutional Neural Networks (CNN) aided by the Grey Wolf optimization algorithm for training, the authors show that a one-dimensional CNN can predict all the studied states (flapping frequency, pitch angle, wind velocity, feather arrangement) with relatively small errors from the real states. Further, the authors show the prediction gives accurate prediction of the states even when done in (near) real time (1s measurement and then immediate evaluation). Finally, the authors demonstrate that the piezoelectric sensors can even generate some power, that can be reused e.g. by powering other on board devices such as other sensors.

In the reviewer's opinion, the work represents a significant step forward in embodied perception and may lead to lighter and smaller flapping-wing robots in the future. Such robots will have limited lifting capabilities and so an embodied sensor that could sense multiple states is highly desirable.

The authors conducted the study systematically and thoroughly, the methodology, data analysis and its interpretation is sound, and overall supports their conclusions and claims (with minor exceptions, which I list further). Enough detail is provided for the work to be reproduced (again, with minor exceptions listed below). I recommend the work for publication once my minor comments below are addressed in the final revised manuscript.

Matej Karasek, Flapper Drones

Response: We sincerely thank you for your positive feedback and valuable suggestions on our research work. Below, we provide detailed responses to each of the comments and recommendations.

Minor comments:

- The claims made about the real-time estimation suggest the CNN can estimate continuous states (e.g. abstract, line 25). However, the results presented show that the data is segmented first (1 s) and then evaluated, so the output will be delayed and downsampled. For feedback control, such delay could cause instability, could the authors discuss why 1s

segments were chosen? Could the authors estimate what would be the minimal segment duration (e.g. in function of the flapping frequency) for this method to work reliably?

Response: Thank you for your comments and suggestions. To illustrate the impact of input sample size on recognition accuracy, we analyzed the identification of flapping frequency using a range of input sample sizes from 0.01s to 10s. We evaluated the effect of different input sample sizes on the recognition error (root mean square error, RMSE) of the CNN network. For each sample size, we conducted five rounds of network training and averaged the RMSEs to ensure the reliability of the results. As shown in the figure below, the results indicate that as the input sample size decreases, the recognition error gradually increases. When the input sample size is 0.01s, the recognition error approaches 0.9Hz, while increasing the input sample size to 1s reduces the recognition error to 0.1Hz. Further increasing the input size shows the error stabilizing, with the recognition accuracy reaching saturation. Therefore, to balance recognition accuracy and reduce network training complexity, we selected 1s as the optimal input sample size.

To address the issue of delayed feedback caused by excessively large input sizes, we propose using a sliding window sampling method rather than reducing the input sample size. Reducing the sample size would lead to a loss of valuable data, thereby decreasing recognition accuracy. Conversely, sliding window sampling can achieve higher sampling rates, thus reducing delay without compromising the input sample size. We investigated the impact of various sliding step sizes on frequency recognition accuracy under temporally varying motion conditions, as shown in Fig. 5d. Using a sliding step size of 1s corresponds to our original approach, where samples are taken every second. We used the same network for recognition across all step sizes, which was trained using our original sampling scheme. With a sliding step size of 1s, the average error was 0.075Hz and the RMSE was 0.1Hz. The comparison between recognized and actual values is shown below. As the sliding step size decreased (increasing the sampling rate), the recognition error slightly decreased and remained stable. At a step size of 0.01s, the average error was 0.072Hz and the RMSE was 0.097Hz, maintaining good recognition accuracy with a sampling rate of 1000Hz. This allows frequency recognition with a time resolution of 0.01s, demonstrating excellent temporal precision and significantly

reducing sampling delay. Notably, a 0.01s step size is not our limit; this approach offers a feasible solution for reducing delay.

We added the discussion of input size and sliding windows step in **Main text** as follows:

(page 11, lines 313-314) “To balance recognition accuracy with the complexity of network training, we choose 1 second as the optimal size (Supplementary Fig. 24) of the input sample.”

(pages 12-13, lines 345-355) “To ensure sufficient temporal resolution during the recognition process, we employed a sliding window approach to segment variable-frequency time-series signals (Fig. 5a). Subsequently, neural networks in Fig. 4g were utilized to identify changes in flapping frequency and generate fitted curves of flapping frequency over time. Results of testing and recognition under a sampling sliding step of 0.01s are shown in Figs. 5d and 5e (Supplementary Movie 6 for data processing details). Comparing results under different sliding step sampling rates (Supplementary Fig. 28) reveals that smaller steps lead to denser recognition points and higher temporal resolution during the process of flapping frequency variation over time. This effect is particularly pronounced in intervals with high rates of flapping frequency change, highlighting the enhanced temporal resolution achieved by reducing the sliding step. Additionally, this approach mitigates output latency introduced by sampling, ensuring accurate and real-time recognition.”

Fig. 5: Experiment of frequency-varying motion signal recognition in continuous time. d Time-history curves of flapping frequency with a sinusoidal variation in continuous time, showing the test values and predicted values (sliding step=0.01s). **e** Time-history curves of flapping frequency with a stepwise variation in continuous time, showing the test values and predicted values (sliding step=0.01s).

We added the discussion of input size and sliding windows step in **supplementary materials** as follows:

(page 38, lines 740-758)“To determine the optimal input sample size, we used a series of input sample sizes from 0.01s to 10s to investigate the effect of sample size on recognition accuracy. We calculate the root mean square error in flapping frequency identification by the Convolutional Neural Network (CNN) under different input sample sizes, in which we conducted 5 network training sessions for each type of input sample size, and took the average of the five root mean square errors as the final root mean square error as shown in the Supplementary Fig. 24 below to ensure the reliability of the results. The findings indicate that as the input sample size decreases, the recognition error gradually increases. when the input sample size is 0.01s, the recognition error approaches 0.9 Hz. Conversely, when the input sample size increases to 1s, the recognition error decreases to 0.1 Hz. Furthermore, as the sample size continues to increase, the recognition error gradually tends to be stable, and the recognition accuracy plateaus. Consequently, to balance recognition accuracy with the complexity of network training, we choose 1s as the size of the input sample.”

Supplementary Fig. 24: Root mean square error (RMSE) in flapping frequency identification across diverse sample input dimensions. The box plots show the maximum, minimum, 25th percentile, 75th percentile, and median error values for each category. Black points represent the mean error for each size.

(page 44, lines 830-836)

Supplementary Fig. 28: Flapping frequency recognition results at different sliding window step sizes. a Sliding window step = 1s, absolute error = 0.0746Hz. **b** Sliding window step = 0.5s, absolute error = 0.0710Hz. **c** Sliding window step = 0.2s, absolute error = 0.0713Hz. **d** Sliding window step = 0.1s, absolute error = 0.0714Hz. **e** Sliding window step = 0.05s, absolute error = 0.0717Hz. **f** Sliding window step = 0.01s, absolute error = 0.0722Hz.

- The authors speak about low weight of their solution (abstract, line 26, and lines 462-3), but I did not manage to find any mass specification of the added sensors. E.g. a mass breakdown table or percentage of the sensor mass to the total wing- and robot mass should be presented. In a free flying robot, such added mass would encompass also all the required circuitry to digitize and process the sensor signals and estimate the states.

Response: Thank you for your comments. In the revised manuscript showing free flying, we developed a untethered flapping-wing aircraft equipped with an integrated signal acquisition and wireless transmission module. The module's circuit board measures 2.2 cm × 1.6 cm × 1.6 mm and weighs 1.465g. The PVDF flexible film weighs 0.225g. We have included a detailed mass breakdown of the entire aircraft in the supplementary materials. It is evident that, for the complete system we developed, the PVDF component accounts for only 0.79% of the total weight, while data acquisition and wireless transmission module constitutes 5.15% of the total weight, with the majority of the mass being attributed to the airframe.

We added the details of flapping robot in **Main text** as follows:

(pages 16-17, lines 455-462) “To substantiate the viability of the bio-hybrid embodied perception approach in practical flight contexts, we meticulously crafted a feathered flapping-wing aircraft (Fig. 8a) capable of untethered flight (Figs. 8b and 8c, Supplementary Movie 11). This aircraft incorporates a gear-driven mechanism, a flight control board, piezoelectric signal acquisition, and high-impedance wireless transmission modules (Supplementary Figs. 33-35). With a total mass of 28.465 g (detailed mass distribution outlined in Supplementary Table 6), the PVDF film represents merely 0.79% of the total mass, whereas the data acquisition and wireless transmission module accounts for 5.15% of the total weight, with the bulk of the mass allocated to the airframe.”

We added the mass decomposition chart in **supplementary materials** as follows:

(page 54, lines 1015-1017) **Supplementary Table 6. Mass decomposition chart for feathered flapping-wing robots**

Compositional parts	Mass(g)	Percentage
PVDF	0.225	0.79%
Signal acquisition and wireless transmission module	1.465	5.15%
Conducting wire	1.834	6.44%
Feathered wings	4.449	15.63%
Fuselage (skeleton + flight control board + power supply)	20.492	71.99%
Total	28.465	100.00%

- The authors emphasize the ability of the piezoelectric sensors to recover energy. To quantify such benefit, could the authors estimate how much power could be recovered, and how it compares e.g. to the power needed to flap the wings? (section at line 530)

Response: Thank you for your comments. As flapping-wing drones trend towards miniaturization, the issue of energy supply emerges as a critical challenge. The payload capacity constraints impose limitations on the carriage of power sources and other electronic components, such as sensors. Our proposed multifunctional biohybrid wing presents a promising solution to this conundrum. It enables sensing without reliance on external power sources while concurrently harvesting embodied energy, thereby enhancing energy utilization efficiency. To quantify the energy harvesting potential of the feather-piezoelectric sensor, we conducted power testing experiments utilizing the flapping apparatus depicted in Fig. 3c and a

resistor box. Under a fixed flapping frequency, the electrical output of the feather-piezoelectric unit was recorded by a data acquisition system at a sampling rate of 1000 Hz to assess the power generation performance of the feather-piezoelectric sensor. The resistor box was employed to measure the optimal impedance of the feather-piezoelectric unit, aiming to explore the maximum output power of the system. The experimental results, as illustrated in Supplementary Fig. 10, indicate that the effective output voltage increases with the load resistance. At an external resistance of 0.8 M Ω , the effective output voltage reached 0.303V, with a power density of 6.226 $\mu\text{W cm}^{-3}$. Although the energy harvested by our flapping device is not substantial, it indeed demonstrates the feasibility of energy harvesting from flapping motion. The flapping energy, after prolonged charging, powers electronic clocks and humidity sensors. Our latest feathered flapping-wing robot can generate a peak open-circuit voltage of 4V and an effective voltage of 1.6V during flapping, significantly surpassing the effective voltage achieved by the aforementioned device, indicating a greater potential for energy recovery. It can serve as a backup emergency power source, sending location information when the primary power source is depleted, aiding in search and rescue operations. Future enhancements in energy harvesting performance can be achieved by increasing the flapping frequency, optimizing the flapping structure, or integrating large deformation joints.

*We added the description of energy harvesting in **Main text** as follows:*

(page 9, lines 241-244) "Although our current energy harvesting efficiency is not remarkable (Supplementary Fig. 10), future improvements can be achieved by optimizing the flapping wing structure and increasing the flapping frequency, providing valuable insights for the energy supply of micro-sized FWRs."

*We added the relevant experimental results in **supplementary materials** as follows:
(page 18, lines 392-396)*

Supplementary Fig. 10: Output performance of the feather-PVDF bio-hybrid sensor. a Output power of the feather-PVDF bio-hybrid sensor under different load resistances. **b** RMS output voltage of the feather-PVDF bio-hybrid sensor under different load resistances.

- It is not entirely clear to the reviewer, whether a single trained CNN can estimate all the studied states (which the reviewer assumes), or whether for each case a separate training was used. Please make this more explicit in the text.

Response: Thank you for your comments. To address the specific requirements of each studied state (flap frequency, wind speed, and pitch angle), we utilized a separate training for each condition. This approach was adopted to optimize the accuracy and reliability of the CNN's estimations for each individual parameter. Detailed information regarding the network architectures and training parameters for each condition can be found in the supplementary materials. Additionally, we have revised the main text to explicitly clarify our methodology.

We added the specific description of the network in **Main text** as follows:

(page 11, lines 309-313)“Three separate 1D CNNs were trained for the extraction of corresponding piezoelectric signal data features and regression identification of the three flight parameters: flapping frequency, wind speed, and pitch angle. We employed GWO to optimize critical parameters for each CNN network (Supplementary Tables 1-3 and Supplementary Movie 6).”

We added the description of the network structure in **supplementary materials** as follows:

(page 39, lines 763-770)“The initial data were normalized and transformed into the input format of $1000 \times 1 \times 1 \times N$ (N is the number of samples) for training. The network structure is illustrated in

Fig. 4g, comprising three convolutional layers, three pooling layers, and three fully connected layers, with the specific stacking sequence as follows: data input → convolutional layer → loss function (ReLU) → pooling layer → convolutional layer → loss function (ReLU) → pooling layer → convolutional layer → loss function (ReLU) → pooling layer → fully connected layers (3 layers) → regression layer.”

(page 40, lines 786-788)Supplementary Table 1. The hyperparameters of the neural network for flapping frequency recognition

No	Layer Type	Kernel size	Kernels number	Pool Size	Stride	FC Size	Learning rate
1	Conv_1	206	143				0.0017
2	Maxpool_1			9	25		
3	Conv_2	231	38				
4	Maxpool_2			10	2		
5	Conv_1	39	24				
6	Maxpool_1			1	1		
7	FC					44	
8	FC					17	
9	FC					1	
10	Regression						

(pages 40-41, lines 789-791)Supplementary Table 2. The hyperparameters of the neural network for oncoming flow velocity recognition

No	Layer Type	Kernel size	Kernels number	Pool Size	Stride	FC Size	Learning rate
1	Conv_1	184	197				6.11e-04
2	Maxpool_1			82	16		
3	Conv_2	148	159				
4	Maxpool_2			13	6		

5	Conv_1	1	206				
6	Maxpool_1			1	32		
7	FC					12	
8	FC					16	
9	FC					1	
10	Regression						

(page 41, lines 792-794) **Supplementary Table 3. The hyperparameters of the neural network for pitch angle recognition**

No	Layer Type	Kernel size	Kernels number	Pool Size	Stride	FC Size	Learning rate
1	Conv_1	145	188				0.0016
2	Maxpool_1			97	13		
3	Conv_2	66	224				
4	Maxpool_2			21	8		
5	Conv_1	48	143				
6	Maxpool_1			2	6		
7	FC					43	
8	FC					45	
9	FC					1	
10	Regression						

- line. 467 the implications to environmental monitoring are not clear to the reviewer, please explain in more detail.

Response: Thank you for your comments. The energy generated during flapping flight has been demonstrated to drive small electronic components, thereby enabling the operation of miniature environmental monitoring sensors (e.g., temperature, humidity, light intensity, UV intensity). This potential application leverages the harvested energy from flapping motion for self-powered environmental sensing [1]. We have also supplemented the main text to provide

clearer understanding for the readers.

[1] Hou, K., Tan, T., Wang, Z., Wang, B. & Yan, Z. Scarab beetle-inspired embodied-energy membranous-wing robot with flapping-collision piezo-mechanoreception and mobile environmental monitoring. *Adv. Funct. Mater.* **34**, 2303745(2024).

We added the description of environmental monitoring in Main text as follows:

(page 9, lines 238-241) “Harvesting energy accumulated from flapping motion can also provide power for portable airborne environmental monitoring modules such as light intensity and ultraviolet radiation sensors⁵¹, enhancing the diversity of environmental parameter monitoring and data collection.”

- Feathered robot design (section at line 573): is this own robot design of the authors, or an adapted off-the-shelve robot? In both cases, please refer to more details about the robot, or report the robot make and model. It seems to be a different robot than the one used in the wind tunnel experiment (Figure 3c)

Response: Thank you for your comments. Our study includes three types of flapping devices: First, the device depicted in Fig. 3c, with detailed design provided in Supplementary Fig. 5. Second, adaptations of a feathered flapping-wing robot shown in Fig. 6a and 6f, derived from the Meta-Bird model (Bionic Bird), where we replaced the wings with our designed feather wings and connection components. This detail has been included in the Supplementary Materials. Third, the feathered flapping-wing robot demonstrated in Fig. 8a, designed for untethered flight, with related fabrication details supplemented in both the main text and Supplementary Materials.

We added the details of the third robot construction in Main text as follows:

(pages 16-17, lines 455-462) “To substantiate the viability of the bio-hybrid embodied perception approach in practical flight contexts, we meticulously crafted a feathered flapping-wing aircraft (Fig. 8a) capable of untethered flight (Figs. 8b and 8c, Supplementary Movie 11). This aircraft incorporates a gear-driven mechanism, a flight control board, piezoelectric signal acquisition, and high-impedance wireless transmission modules (Supplementary Figs. 33-35). With a total mass of

28.465 g (detailed mass distribution outlined in Supplementary Table 6), the PVDF film represents merely 0.79% of the total mass, whereas the data acquisition and wireless transmission module accounts for 5.15% of the total weight, with the bulk of the mass allocated to the airframe.”

Fig. 8: Flight testing of the flapping-wing robot with feathered wings. a Structural components of the flapping-wing robot with feathered wings.

We added the details of the second robot construction in *supplementary materials* as follows: (page 45, lines 842-849)“This flapping-wing robot is adapted from the Meta-Bird model by Bionic Bird. We removed its original wings and installed custom-designed, 3D-printed attachments on both sides. One of these attachments is designed to accommodate the insertion of six feathers.”

Supplementary Fig. 29: Display and parameter description of the feathered flapping wing robot. a Dimensions of the Feathered Flapping Wing Robot. **b** Weight of the Feathered Flapping Wing Robot.

We added the details of the third robot construction in *supplementary materials* as follows: (pages 52-54, lines 976-1014)“We have developed a customized ornithopter featuring a wing-flapping mechanism. The wings are constructed with a PET film as the primary support layer, with the PVDF-feather hybrid sensor mounted on the outermost edge of the wing. Additional feathers are layered sequentially over the PET base. Lightweight clasps are positioned along the wing's central axis to ensure secure attachment and stability during flapping. Wing motion is driven by a

gear assembly and a miniature motor.

A signal acquisition/wireless transmission module, measuring 2.2cm × 1.6cm × 1.6mm and weighing 1.465g, is mounted at the rear of the ornithopter and connected to the PVDF-feather hybrid sensors. Both the flight control board and the signal acquisition/wireless transmission module are powered by a single 1s lithium battery, with the entire robot weighing 28.465g. This setup enables untethered flight, as well as signal acquisition and wireless transmission. The overall operational principle is illustrated in the Supplementary Fig. 33. The first part is the flight drive module, which includes the flight control board, micro motors, and gear set to control the wing flapping frequency. The specific component connection diagram is shown in Supplementary Fig. 34. The electronic components on the flight control board include an MCU (PAN742), IMU (BMI270), Low-Dropout Regulator, and a wireless transceiver module (CC2530), all powered by a 100mAh 1S lithium battery. The second part is the biohybrid sensing module, which comprises the biohybrid sensor, signal acquisition/wireless transmission circuit board, and CNN algorithm to enable the recognition of flight parameters. This module is also powered by the same 100mAh 1S lithium battery. The specific component connection diagram is shown in Supplementary Fig. 35. The electronic components on the signal acquisition/wireless transmission board include an MCU (STM32), Low-Dropout Regulator, a wireless transceiver module (CH9140), operational amplifiers, and switched capacitor inverter. The sensor is connected to the signal acquisition/wireless transmission board via wires. Bluetooth communication is achieved using BleComManager software, with the host computer sending commands for data acquisition and wireless transmission. The sensor data is then processed by the CNN algorithm module for recognition.”

Supplementary Fig. 33: Operational principles of the flapping-wing robot.

Supplementary Fig. 34: Schematic diagram of the component connections in the drive module

Supplementary Fig. 35: Integrated signal acquisition and wireless transmission module. a Display of modules. **b** Schematic diagram of the component connections in the perception module.

- Can you give more details about the wind tunnel type, and possibly in which facility it is located?

Response: Thank you for your comments. The wind tunnel is located at the Engineering Mechanics Laboratory of Shanghai Jiao Tong University. It is an open-circuit, straight-through wind tunnel with a total length of 4.71 meters and a width of 1.02 meters. The tunnel comprises an expansion section, a stabilizing section, a contraction section, a test section, and a tailpipe. It includes six layers of honeycomb grids and one layer of honeycomb flow straighteners. The wind is driven by a variable frequency controlled three-phase asynchronous motor rated at 7.5

kW with a nominal voltage of 380V. Specific details about the wind tunnel have been added to the Supplementary Materials.

*We added the details of the wind tunnel in **supplementary materials** as follows:*

(page 16, lines 375-384)“The wind tunnel, located at the Engineering Mechanics Experiment Center of Shanghai Jiao Tong University, is an open-circuit type with a total length of 4.71 m and a width of 1.02 m. It comprises an expansion section, settling chamber, contraction section, test section, and exhaust duct. The test section is 0.60 m in length with a square cross-section having a side length of 0.33 m and an inner wall thickness of 15 mm. The top features a removable rectangular cover plate measuring 0.46 m in length and 0.24 m in width. The tunnel features six layers of honeycomb and mesh screens for flow straightening and is powered by a variable-frequency, three-phase asynchronous motor with a rated power of 7.5 kW and a rated voltage of 380V.”

Reviewer #1 (Remarks to the Author):

I appreciate the effort the authors have made to reply my previous comments. I feel that the quality of the paper has been improved to some extent after revision. However, the concern is the major issues raised in the previous version have not been fully addressed and the reviewer is still not convinced about the novelty of the sensor and signal processing aspect. This seems to be a good application, but only a good application and a stack of trivial work and analysis cannot elevate the key novelty of the work.

Response: Thanks for your valuable suggestions and insights regarding our work, which have prompted us to focus more intently on enhancing the quality and novelty of our manuscript. Biological wings are equipped with mechanoreceptors that can detect complex aerodynamic loads, enabling immediate proprioceptive flight control. Currently, inertial sensors embedded within the rigid body struggle to manage the intricate aerodynamic behaviors associated with flexible flapping wings, thus limiting the feedback control of flapping vehicles. Notably, there are no existing examples of untethered flapping vehicles achieving precise embodied perception. Inspired by birds, we propose a biohybrid flapping vehicle that achieves precise proprioceptive recognition for untethered flight through embodied perception of the feather-PVDF mechanoreceptor, providing a crucial conceptual framework for the embodied intelligence design of flapping vehicles. Finally, we would like to express our sincere gratitude once again for the time and effort you have dedicated to our manuscript.

Reviewer #2 (Remarks to the Author):

I appreciate the authors taking the time to re-develop hardware to respond in earnest to all of the reviewer feedback. The new direction, supported by the development of novel hardware, on which their innovation can be tested, is a significant achievement in the field. As a result, the work is suitable for publication in this journal. I see no other errors that need correcting, and the new message for the work is significantly more compelling - the work is ready for publication without any additional changes.

Reviewer #2 (Remarks on code availability):

N/A

Response: We sincerely appreciate the valuable suggestions you provided regarding our work. Your insights have greatly assisted us in clarifying our direction and significantly enhancing the quality of the manuscript. Following your suggestion, we provide the code in the final version of our submission. Thank you once again for the time and effort you invested, as well as for your recognition of our efforts.

Reviewer #3 (Remarks to the Author):

The authors have answered all my comments to my full satisfaction and improved the

manuscript accordingly; the limitations of the methodology and applicability of the results to FWMAVs got further clarified. The newly added flight experiments and other details added in response to the other reviewers' comments, further strengthen the scientific contribution of the manuscript.

I recommend the paper for publication.

I also recommend publishing all the raw experimental data as a dataset, such that interested readers can duplicate the results following the published methodology.

Reviewer #3 (Remarks on code availability):

No code provided

Response: We are immensely grateful for your affirmation of our work, which has provided us with considerable confidence. We also sincerely appreciate your invaluable suggestions, which have further refined our research and enhanced its scientific rigor—an aspect that is critically important to us. We include the relevant original experimental data and code in the final version of our submission. Finally, we thank you once again for the time and effort you have devoted to our manuscript.